# SimBench: Benchmarking the Ability of Large Language Models to Simulate Human Behaviors

**Tiancheng Hu**[1], **Joachim Baumann**[2], **Lorenzo Lupo**[3],
**Nigel Collier**[1], **Dirk Hovy**[3], and **Paul Röttger**[4]

[1]University of Cambridge, [2]Stanford University, [3]Bocconi University, [4]University of Oxford
th656@cam.ac.uk

🤗 Data    ⬡ Code    🌐 Website

## Abstract

Large language model (LLM) simulations of human behavior have the potential to revolutionize the social and behavioral sciences, *if and only if* they faithfully reflect real human behaviors. Current evaluations of simulation fidelity are fragmented, based on bespoke tasks and metrics, creating a patchwork of incomparable results. To address this, we introduce SimBench, the first large-scale, standardized benchmark for a robust, reproducible science of LLM simulation. By unifying 20 diverse datasets covering tasks from moral decision-making to economic choice across a large global participant pool, SimBench provides the necessary foundation to ask fundamental questions about when, how, and why LLM simulations succeed or fail. We show that the best LLMs today achieve meaningful but modest simulation fidelity (score: 40.80/100), with performance scaling log-linearly with model size but not with increased inference-time compute. We discover an alignment-simulation tradeoff: instruction tuning improves performance on low-entropy (consensus) questions but degrades it on high-entropy (diverse) ones. Models particularly struggle when simulating specific demographic groups. Finally, we demonstrate that simulation ability correlates most strongly with knowledge-intensive reasoning (MMLU-Pro, $r = 0.939$). By making progress measurable, we aim to accelerate the development of more faithful LLM simulators.

We combine **20 datasets** in a unified format.

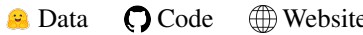

| | |
|---|---|
| ChaosNLI | MoralMachineC |
| Choices13k | AfroBarometer |
| OpinionQA | OSPsychBig5 |
| NumberGame | DICES990 |
| WisdomOfCrowds | Jester |
| LatinoBarometro | ISSP … |

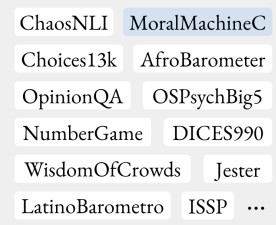

A train will kill 5 people on the track. You can flip a switch to divert the train to a side track where it will kill just 2 people.

**What do you do?**
  **A**:  Flip the switch
  **B**:  Do nothing

Each dataset contains **multiple-choice questions**.

We test the ability of LLMs to simulate **group-level responses**.

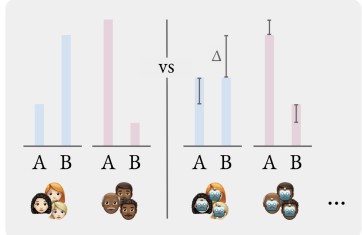

Figure 1: SimBench is the first large-scale benchmark to evaluate how well LLMs can simulate group-level human behavior across diverse simulation settings and tasks.

## 1 Introduction

Large-scale human experiments and surveys have long been essential tools for informing public policy, commercial decisions, and academic research. Running experiments and surveys, however, is costly

and time-consuming. Large language models (LLMs) may help address this challenge by simulating human behaviors quickly and at low cost, to complement or even substitute human studies. This prospect, alongside encouraging early evidence on the efficacy of LLMs as simulators (Aher et al., 2023; Argyle et al., 2023; Horton, 2023), has motivated a large body of recent work across many disciplines investigating the ability of LLMs to simulate human behaviors (Bisbee et al., 2024; Dominguez-Olmedo et al., 2024; Manning et al., 2024; Binz et al., 2025; Hu & Collier, 2025, inter alia).

However, this rapid exploration has produced a fragmented body of evidence. Most studies evaluate a narrow set of LLMs on a specific task, yielding varied and sometimes contradictory results that make it difficult to draw broader conclusions (§5). The field lacks a unified framework to determine when, how, and why LLM simulations succeed, or how to train better simulators.

To confront this challenge, we introduce SIMBENCH: the first large-scale, standardized benchmark for group-level human behavior simulation. By harmonizing 20 diverse datasets, which span moral dilemmas, economic games, and psychological assessments across a vast global participant pool, SIMBENCH enables rigorous measurement and comparison of simulation fidelity across models, tasks, and populations.

Using SIMBENCH, we move beyond isolated experiments to build a comprehensive picture of LLM simulation. We structure our investigation around six research questions. We first establish a baseline, asking **how well current LLMs perform at social simulation** (RQ1) and **how characteristics like model size and inference-time compute affect their simulation ability** (RQ2). We find that top models achieve meaningful but modest simulation fidelity (top score: 40.80/100), though performance scales log-linearly with size and, surprisingly, is not improved by scaling inference-time compute. Next, we explore the sources of this variance, asking **how task selection** (RQ3) and **human response plurality** (RQ4) affect simulation fidelity. We find that fidelity varies substantially by task, and uncover an alignment-simulation tradeoff: instruction tuning's mode-seeking objective systematically improves performance on low-entropy (consensus) questions but harms performance on high-entropy (pluralistic) questions. A causal analysis confirms this tradeoff stems from two opposing forces: a beneficial instruction-following effect and a harmful entropy-reduction effect. Finally, we ask if LLMs are **better at simulating some demographic groups than others** (RQ5) and **to what extent simulation ability correlates with other model capabilities** (RQ6). We show models struggle most with religious/ideological groups and that simulation ability correlates most strongly with deep, knowledge-intensive reasoning (MMLU-Pro, $r = 0.939$).

Progress in AI is only possible through rigorous evaluation, and large-scale benchmarks such as MMLU (Hendrycks et al., 2021) have significantly contributed to improvements in LLM capabilities. In the same spirit, SIMBENCH provides the foundational infrastructure needed to move LLM simulation from a collection of ad-hoc studies to a measurable and systematic science. We note that while predicting group-level response distributions is the instrument used here for benchmarking, we view this as a foundational proxy for the broader capability of human behavior simulation. We acknowledge that full human simulation inherently includes interactive and complex dynamics not fully captured by static response distributions (see Appendix A). We have a Project Website, and all of SIMBENCH is available on GitHub and HuggingFace.

## 2 CREATING SIMBENCH

### 2.1 DATA CURATION

To create SIMBENCH, we combine a repository-driven approach, where we query major social and behavioral science repositories (e.g., Harvard Dataverse, ICPSR, OSF), and a literature-driven approach, where we identify key papers in relevant fields and trace back to their underlying data sources. We then apply a strict set of selection criteria to all candidate datasets: **large participant counts**, to ensure meaningful group-level distributions; **permissive licensing** to allow for redistribution; **single-turn, self-contained questions**, to establish a standardized evaluation paradigm free from multi-turn or contingent interactions; **multiple-choice or ordinal response formats**, to enable quantitative evaluation; and **English-language questions** or validated translations for consistency.

We complement these criteria with a curation strategy that balances competing objectives. We prioritize **novelty**, favoring datasets not previously used in LLM simulation evaluation, but also

ensure **backward comparability** by including well-established benchmarks (e.g., OpinionQA, ChaosNLI). Furthermore, we prioritize datasets with rich **sociodemographic data** to enable fine-grained analysis of specific subpopulations (§2.4). However, we make targeted exceptions for datasets like Jester and Choices13k, which, despite lacking demographic data, provide unique and essential task diversity.

This principled curation process yields the **20** datasets that comprise SIMBENCH, which we list in Appendix L, providing details on participants and example questions. To demonstrate the rigor of our process, and as a resource for the community, we list datasets that were considered but ultimately excluded in Appendix B. Crucially, SIMBENCH is fully modular by design, so that future work can easily add more datasets using the processing pipeline described in §2.3 below.

## 2.2 BENCHMARK PROPERTIES

As a whole, SIMBENCH is defined by two key properties: task diversity and participant diversity.

1) **Task Diversity**: SIMBENCH tasks are highly diverse in terms of the human behavior they capture. SIMBENCH includes **decision-making** questions (e.g., in Choices13k, MoralMachine), where participants are presented with a set of actions that concern themselves, and they have to select the action they would hypothetically take. SIMBENCH also includes **self-assessment** questions (e.g., in OpinionQA, OSPsychBig5), where participants are presented with a set of descriptions or attributes, and they have to select the one that best describes themselves. Further, SIMBENCH includes **judgment** questions (e.g., in ChaosNLI and Jester) where participants are presented with some external object and a choice of labels, and they have to select the label they think fits best. Lastly, SIMBENCH includes **problem-solving** questions (e.g., in WisdomOfCrowds and OSPsychMGKT), where participants are presented with a set of answers to a factual question, and they have to select the answer they think is correct. Consequently, LLMs have to accurately simulate several distinct types of human behavior in order to perform well on SIMBENCH.

2) **Participant Diversity**: SIMBENCH captures a rich demographic landscape spanning more than 130 different countries across six continents (see Appendix O for a full country-level breakdown). SIMBENCH prioritizes international representation: samples from the Anglosphere West constitute only 27.9% of the data.[1] This substantial global scope is driven by a diverse collection of sources: 3 datasets (e.g., Latinobarómetro, Afrobarometer) exclusively feature participants from regions outside the US, 4 datasets (e.g., GlobalOpinionQA, TISP) draw from multi-country samples across different continents, and 2 datasets collect responses from a global pool of internet users. Importantly, 8 out of the 20 datasets employ representative sampling techniques, enhancing the ecological validity of these constituent components. To perform well on SIMBENCH, LLMs must therefore demonstrate the ability to accurately simulate the behavior of human participants across diverse cultural, linguistic, and socioeconomic backgrounds.[2]

## 2.3 UNIFYING SIMBENCH DATASET FORMATS

A core contribution of SIMBENCH is the harmonization of 20 heterogeneous datasets into one standardized format. This process ensures that LLM simulation ability can be compared rigorously across diverse tasks and populations.

**Question Normalization:** We standardize all items into a multiple-choice format, making minimal edits that primarily consist of mapping existing discrete options to standardized letter keys, each corresponding to a single token, to enable clean extraction of per-option probabilities from base models while preserving the original experimental structure. For the few datasets with continuous scales (Jester), we map responses to discrete bins. We further ensure consistency by collapsing answer options where appropriate, limiting the maximum to 26 choices (though typically fewer than

---

[1]We define the Anglosphere West as the U.S., Canada, U.K., Australia, and New Zealand. Even using a broader definition of "the West" that includes Western Europe, these nations account for less than half (45.9%) of the benchmark.

[2]Note that, while some constituent datasets recruit representative samples, SIMBENCH as a whole is not fully representative of any single population. We discuss this limitation in Appendix A.

10), and using the official English-language versions of all questions.[3] This is a deliberate choice to standardize the evaluation and avoid confounding simulation ability with multilingual performance capabilities, ensuring that differences in scores reflect simulation fidelity rather than translation quality, even for datasets originally collected in local languages.

**Response Aggregation:** To evaluate group-level simulation, we standardize all data into group-level probability distributions. For the majority of our datasets, which provide raw individual-level responses, we create these distributions by aggregating the data ourselves. Post-stratification weights are applied whenever applicable (e.g., ESS). For the few datasets that are already provided in an aggregated format (e.g., GlobalOpinionQA), we process and normalize their existing statistics to conform to the SIMBENCH schema.

We create the simulation targets — the ground-truth response distributions that models must predict — in two ways: 1) **Default Grouping**. For every question in a dataset, we create a baseline target by aggregating responses from all participants. This represents the "default" population for that dataset (e.g., "US-based Amazon Mechanical Turk workers") and is used to measure general simulation ability. 2) **Specific Grouping**. For datasets with rich sociodemographic data, we create more fine-grained targets by aggregating responses from participants sharing a specific attribute (e.g., age or gender). This allows us to evaluate LLM ability to simulate narrower, more specific demographic groups. We detail the available grouping variables for each dataset in Appendix L.

Each simulation target is paired with a prompt that describes the corresponding group. Overall, this harmonization process yields **10,930,271** unique question-group simulation targets. From this set, we curate our final benchmark splits (§2.4) to enable robust evaluation of LLM simulation capabilities. We note that while training data contamination remains an inherent risk, our zero-shot aggregation task largely mitigates this by testing distributional prediction (see Appendix A for a full discussion).

## 2.4 SIMBENCH SPLITS

The full set of over 10 million simulation targets is too vast for practical evaluation. We therefore curate two benchmark splits, each designed to probe a different facet of LLM simulation ability.

1) The **SimBenchPop** split covers all questions in all 20 datasets after processing as in §2.3. We combine each question with the dataset-specific default grouping prompt to create one unique test case, resulting in 7,167 test cases. We obtain the response distribution for each test case by aggregating all individual responses to that test case over all participants in that dataset. Conceptually, **SimBenchPop measures the ability of LLMs to simulate responses of broad and diverse human populations**.

2) The **SimBenchGrouped** split draws from the five large-scale survey datasets in SIMBENCH (Afrobarometer, ESS, ISSP, Latinobarómetro, and OpinionQA), which are the only datasets with sufficient participant counts to yield reliable response distributions even when conditioning on a specific demographic attribute (e.g., age = 30-49). For each dataset, we select questions that exhibit significant variation across demographic groups, ensuring that the benchmark captures meaningful demographic differences in responses. This results in 6,343 test cases overall. For more details on the sampling process, see Appendix C. Conceptually, **SimBenchGrouped measures the ability of LLMs to simulate responses from narrower participant groups based on specified group characteristics**.[4]

## 3 EXPERIMENTAL SETUP

**Tested Models:** To demonstrate the utility of SIMBENCH and answer our six research questions (§1), we evaluate 45 recent LLMs. This includes both commercial and open-weight, base and instruction-tuned models, with sizes ranging from 0.5B to 405B parameters. Table 7 lists all models.

---

[3]We note that simulation ability may plausibly be correlated with prompt language, and encourage future work in this direction.

[4]Ideally, we would also like to measure LLM simulation ability for intersectional groups that combine multiple characteristics (e.g., female + age 30-49). However, selecting on multiple characteristics substantially decreases group size, thus increasing sampling noise in the response distributions. Reliable evaluation of intersectional group simulation ability would require datasets with more participants than we have access to.

**Model Elicitation:** For each model, we collect predictions for the two main splits of SIMBENCH (§2.4). To obtain model response distributions, we use one of two methods, depending on model type: 1) For base models, we directly extract **token probabilities** for each response option based on first-token logits. This is a natural way of eliciting a distribution out of an LLM, especially a base LLM. 2) For instruction-tuned models, we follow recent literature on LLM calibration and distribution prediction (Tian et al., 2023; Wang et al., 2024; Meister et al., 2025) and use **verbalized distributions** (e.g., "Option A: 30%, Option B: 70%") elicited through prompting. We empirically validate this methodological choice in Appendix E, which provides strong evidence that verbalized distributions substantially and consistently outperform direct token probabilities for instruction-tuned models. This ensures each model class is evaluated using its most suitable answer elicitation method. For implementation details and prompt formats, see Appendix D.

**Evaluation Metric**: To measure LLM simulation ability, we derive the SIMBENCH score $S$ from the Total Variation Distance (TVD). Conceptually, $S$ quantifies the improvement of a model's prediction $Q$ over a uniform baseline $U$, relative to the human ground truth $P$:

$$S(P,Q) = 100 \left(1 - \frac{\text{TVD}(P,Q)}{\text{TVD}(P,U)}\right) \qquad (1)$$

where a score of 100 indicates perfect alignment and 0 indicates performance equivalent to random guessing. Since in practice a direct point-wise calculation is undefined when the question-level human distribution is uniform ($P = U$), to ensure numerical stability, we compute the score $S_i$ for each specific test case $i$ by normalizing against the *dataset-level* mean baseline:

$$S_i = 100 \left(1 - \frac{\text{TVD}(P_i, Q_i)}{\frac{1}{|D|} \sum_{j \in D} \text{TVD}(P_j, U_j)}\right) \qquad (2)$$

where the denominator represents the average TVD between human responses and the uniform distribution across all test cases $j$ in the dataset $D$. This ensures the metric remains robust across datasets with varying entropy. The final reported score for a model is the average of $S_i$ across all evaluated test cases. As described in §2.4, we cap each dataset at 500 questions to limit over-representation of larger datasets; datasets with fewer total questions naturally contribute fewer test cases and thus carry proportionally less weight in the overall score.

## 4 RESULTS

### 4.1 RQ1:
GENERAL SIMULATION ABILITY OF LLMS

To evaluate general simulation ability, we measure overall SIMBENCH score $S$ averaged across the two main splits of SIMBENCH in Table 1 and Appendix Table 7.

We find that **the best LLMs achieve meaningful but modest simulation fidelity**, as measured

Table 1: **Overall simulation ability of representative LLMs** as measured by SIMBENCH score $S$ averaged across the two main splits of SIMBENCH. Reasoning models are highlighted in *italics*. A full table with all 45 models is in Appendix Table 7.

| Model | Type | Release | $S$ ($\uparrow$) |
|---|---|---|---|
| *Top-Performing Models* | | | |
| Claude-3.7-Sonnet | Instr. | Closed | 40.80 |
| *Claude-3.7-Sonnet-4000* | Instr. | Closed | 39.46 |
| GPT-4.1 | Instr. | Closed | 34.55 |
| *DeepSeek-R1* | Instr. | Open | 34.52 |
| *o4-mini-high* | Instr. | Closed | 28.99 |
| Llama-3.1-405B-Instruct | Instr. | Open | 28.40 |
| Qwen2.5-72B-Instruct | Instr. | Open | 27.61 |
| Qwen2.5-32B-Instruct | Instr. | Open | 23.76 |
| OLMo-2-32B-DPO | Instr. | Open | 19.80 |
| *Top-Performing Base Models* | | | |
| OLMo-2-32B | Base | Open | 15.90 |
| OLMo-2-13B | Base | Open | 13.83 |
| Qwen2.5-72B | Base | Open | 13.34 |
| Qwen2.5-32B | Base | Open | 12.27 |
| *Models Performing Below Uniform Baseline* | | | |
| Gemma-3-4B-PT | Base | Open | -0.65 |
| Qwen2.5-3B-Instruct | Instr. | Open | -12.04 |
| OLMo-2-7B-Instruct | Instr. | Open | -21.36 |

across the 20 datasets in SIMBENCH. Claude-3.7-Sonnet is the best-performing model overall, achieving a score of 40.80 out of 100 on SIMBENCH. While this score means the best model's predictions remain closer to a uniform distribution than to the human ground truth, it nonetheless closes roughly 40% of the gap between the two, demonstrating a genuine and non-trivial simulation signal. The top open-weight model, DeepSeek-R1, scores 34.52. The majority of the 45 models we test perform substantially worse, scoring less than 20. Notably, ten models we test score below 0, indicating that their predicted response distributions are, on average, even further away from the true human response distribution than a uniform response distribution. Statistical analysis, detailed in Appendix H,

confirms that the performance differences between most top-ranked models as well as within each model family are statistically significant. Overall, these results consolidate the mixed findings from prior work into a clearer, if somewhat sobering, picture. When evaluated across a diverse range of tasks and populations, today's LLMs do possess genuine, non-trivial simulation abilities but are still far from being consistently reliable, general-purpose simulators. The stark performance differences between models also caution strongly against the use of smaller, less capable models for simulation, many of which perform worse than a simple uniform baseline.

### 4.2 RQ2: IMPACT OF LLM CHARACTERISTICS ON SIMULATION ABILITY

While even the best models struggle to perform well on SIMBENCH, Table 1 also shows clear differences across models. Therefore, we investigate how performance varies depending on model characteristics, specifically 1) model size, and 2) inference-time compute.

**1) Model Size**   To evaluate the impact of model size on simulation ability, we plot SIMBENCH Score $S$ against model parameter count for the four LLM families that we can test across multiple model sizes (Figure 2 and Appendix Figure 6). Our results suggest that there is a log-linear scaling trend for LLM simulation ability. Across examined model families, for both base and instruction models, an increase in parameter count generally corresponds to an increase in SIMBENCH score $S$, indicating better alignment between predicted and human response distributions. This relationship is robustly supported by model families with comprehensive size variants (e.g., Qwen2.5, Llama-3.1), though additional data points would be needed to fully characterize the trajectory for families with fewer models (e.g., OLMo). There is also an interaction between model size and the effect of instruction tuning. While instruction-tuned models consistently outperform their base counterparts at larger scales (>10B parameters), this relationship appears to invert for smaller models. For example, the OLMo-2 base models outperform their instruction-tuned variants at the 7B and 13B scale. Furthermore, the plot shows that **instruction-tuned models not only reach a higher peak performance but also appear to scale more effectively**. The steeper slope of the dashed lines (e.g., for Qwen2.5-Instruct) compared to the solid lines suggests that instruction tuning may improve a model's ability to capitalize on increases in parameter count for the simulation task. We present a more comprehensive plot including all evaluated model families in Appendix G, which confirms this trend holds across models.

Overall, the clear positive scaling trends across model families suggest that simulation ability is a capability that improves with scale. However, the log-linear nature of this relationship implies diminishing returns from scale alone, reinforcing the need for targeted approaches — such as distribution-preserving alignment (§4.4) — to achieve substantial further gains.

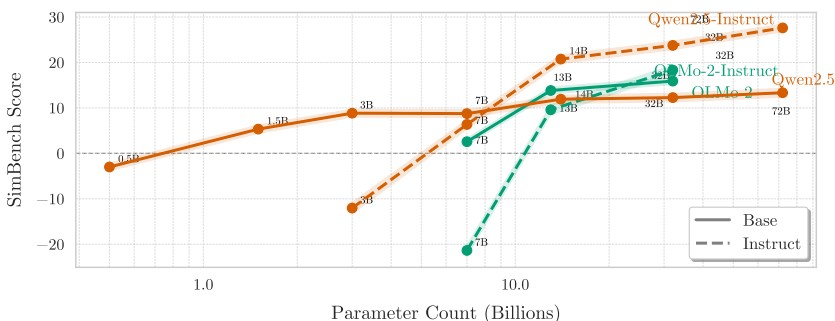

Figure 2: **Model parameter count vs. simulation ability**. We measure model size by parameter count and simulation ability by SIMBENCH score $S$ averaged across the two main splits of SIMBENCH. Shaded bands denote 95% confidence intervals around the mean score.

**2) Inference-Time Compute**   To analyze the effects of increasing inference-time compute on LLM simulation ability, we conduct two sets of experiments. First, we compare the performance of two distinct o4-mini checkpoints ("low" vs. "high", which vary in the amount of reasoning effort), as well as Claude-3.7-Sonnet without reasoning and with a 4000-token reasoning budget. Second, we

apply a zero-shot Chain-of-Thought (CoT) prompting strategy (Wei et al., 2022) to GPT-4.1 and DeepSeek-V3-0324 (see Appendix D for prompt details).

Our results suggest that, across the models tested, **increasing inference-time compute provides no meaningful benefit for LLM simulation ability.** The o4-mini model shows minor improvement (SIMBENCH $S$ score: 27.77 → 28.99), while Claude-3.7-Sonnet's performance slightly decreases (40.80 → 39.46). Similarly, applying CoT prompting leads to a small performance drop for GPT-4.1 (34.55 → 33.11) and a negligible change for DeepSeek-V3-0324 (32.89 → 33.16). While our analysis covers a limited number of models, this result aligns with a growing body of recent work showing that inference-time compute benefits are highly task-dependent (Liu et al., 2025b; Sprague et al., 2025; Gema et al., 2025) and do not necessarily improve role-playing ability (Feng et al., 2025). We hypothesize this is because CoT forces an overly rational deliberation that mismatches the often heuristic-driven nature of human responses in SimBench.

### 4.3 RQ3: IMPACT OF TASK SELECTION ON SIMULATION FIDELITY

The 20 datasets in SIMBENCH correspond to tasks that are highly diverse in terms of which aspects of human behavior they measure (see §2.1). Breaking down simulation fidelity by dataset (Figure 3) reveals that **simulation fidelity varies substantially across tasks**. Models are most successful at simulating responses to standard survey questions regarding stated opinions, attitudes, and self-assessments (e.g., OpinionQA, Afrobarometer). However, performance degrades on tasks requiring the simulation of a *behavioral choice*, whether in risky choice problems (Choices13k) or moral dilemmas (MoralMachine). This finding provides large-scale evidence for a "value-action gap" in LLMs, echoing recent work (Shen et al., 2025) which suggests a misalignment exists between LLM-generated value statements and their actions.

Finally, models struggle severely to capture human response distributions on datasets involving traits or beliefs that conflict with standard alignment objectives. For even the best LLMs, we observe extremely poor performance, often worse than a uniform baseline, on datasets measuring Machiavellianism (OSPsychMACH), conspiratorial beliefs (ConspiracyCorr), or humor rating (Jester). This aligns with a growing body of work (Liu et al., 2024; Kumar et al., 2025; Yi et al., 2025) showing that alignment filters may inhibit the simulation of "atypical" or counter-normative human perspectives. While most LLMs exhibit these performance patterns, GPT-4.1 is a notable outlier, scoring exceptionally high (61.9) on OSPsychRWAS.

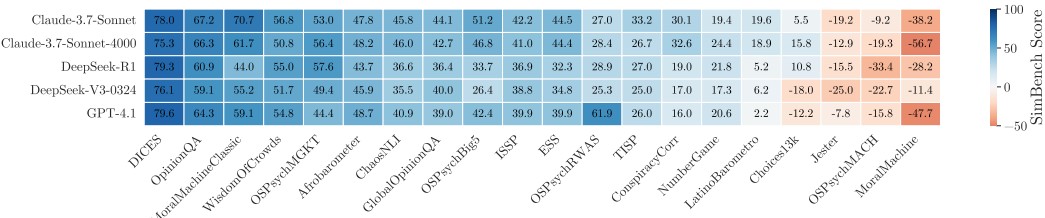

Figure 3: **Simulation fidelity by dataset** as measured by SIMBENCH score $S$ for each of the 20 datasets in SimBenchPop. We show results for the top five models based on results in Table 1.

### 4.4 RQ4: THE ALIGNMENT-SIMULATION TRADEOFF

Faithful simulation requires models to capture the full spectrum of human opinion, from strong consensus to widespread disagreement. We operationalize this "response plurality" using the normalized entropy of the human response distribution.

**An Empirical Tradeoff between Alignment and Plurality.** Prior work suggests that standard alignment via instruction tuning encourages confident, low-entropy outputs (Brown et al., 2020; Tian et al., 2023; Cruz et al., 2024; Meister et al., 2025), which creates a potential conflict with simulating diverse human perspectives. Our analysis reveals exactly this tradeoff. As detailed in Appendix I, base models consistently outperform their instruction-tuned counterparts on high-entropy questions, while the inverse is true for low-entropy questions. To precisely quantify this effect, we compute the

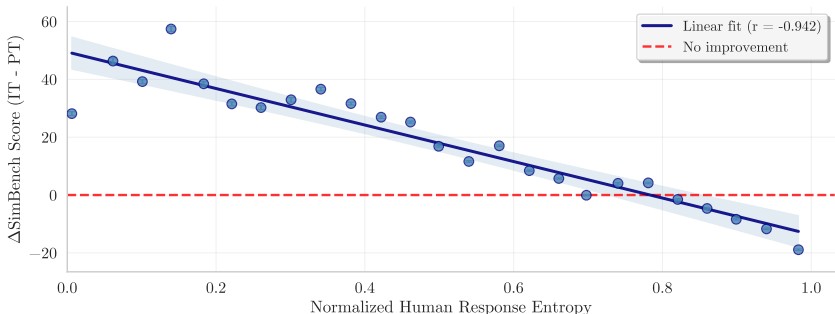

Figure 4: **The Alignment-Simulation Tradeoff: Instruction Tuning Helps on Consensus Questions but Hurts on Diverse ones.** We bin questions by the normalized entropy of their human response distribution (x-axis) and plot the mean improvement in SIMBENCH Score when switching from a base model to its instruction-tuned counterpart (y-axis).

change in SIMBENCH score ($\Delta S = S_{\text{instruct}} - S_{\text{base}}$) for 13 model pairs and plot this performance gain from post-training against human response entropy.

The result, shown in Figure 4, is a near-perfect negative linear relationship ($r = -0.942$). This plot reveals two distinct regimes. On low-entropy questions where humans agree, post-training provides a substantial benefit, improving the S-score by up to 40 points. However, as human disagreement (entropy) increases, the benefit of instruction tuning systematically erodes, crossing a point of no-improvement around an entropy of 0.8. For questions with the highest plurality, post-training becomes actively detrimental, making the aligned model a *worse* simulator than its base counterpart.

This empirical finding is well-explained by the theoretical framework of reinforcement learning (RL) as Bayesian inference (Levine, 2018; Korbak et al., 2022). The pre-training objective of a base model typically minimizes a *mass-covering* KL divergence ($D_{KL}(p\|q)$), which encourages the model ($q$) to place probability mass wherever the true data distribution ($p$) has mass. This process naturally leads to models that represent the full, multi-modal diversity of human language and opinion seen in their training data. In contrast, alignment via KL-regularized RL (e.g., RLHF) minimizes a *mode-seeking* KL divergence ($D_{KL}(q\|\sigma)$). This objective incentivizes the model ($q$) to find and concentrate its probability mass on a single, high-reward mode of the target preference distribution ($\sigma$), even at the cost of ignoring other valid modes. Our results provide strong empirical validation of this theoretical distinction: alignment optimizes for a single "best" response, effectively encouraging the model to discard the pluralistic, high-entropy distributions characteristic of genuine human populations.

To formally test this mechanism, a causal mediation analysis (Appendix I.3) decomposes the effect of instruction tuning into two larger, opposing forces: a large, positive *direct effect* on performance (**+6.46**), likely from improved instruction-following, and a significant, negative *indirect effect* (**-1.74**) mediated by the model's reduced output entropy.

**Case Study: General-Purpose vs. Specialist Cognitive Tuning.** This formally-decomposed tradeoff is perfectly illustrated by comparing general-purpose instruction tuning with the specialist cognitive tuning of the Centaur models (Binz et al., 2025). Centaur models are Llama models fine-tuned on Psych-101, a large dataset of lab experiments, making our diverse SIMBENCH a powerful out-of-distribution test of their generalization. Both approaches improve simulation over the Llama-3.1-70B base model, but they do so via opposing mechanisms. **General-purpose instruction tuning** ($S = 16.56$) leverages the helpful direct effect of alignment, excelling on low-entropy consensus tasks. In contrast, **specialist cognitive tuning** ($S = 8.54$) improves performance by avoiding the harmful indirect effect, preserving the base model's intrinsic ability to capture high-entropy pluralistic responses. The existence of these two distinct—and currently separate—paths to improving simulation underscores a key challenge and opportunity: the most faithful simulators of the future will likely need to synthesize the benefits of both general-purpose alignment and distribution-preserving cognitive modeling. Other promising directions include post-hoc weight interpolation (Hu et al., 2025) and inference-time methods to amplify system prompt adherence beyond the default assistant persona (Dong et al., 2026).

## 4.5 RQ5: SIMULATION ABILITY ACROSS PARTICIPANT GROUPS

Many applications require simulating responses from specific demographic groups rather than general populations. Using SimBenchGrouped, we evaluate how LLM simulation ability changes when conditioned on specific demographic attributes.

We measure this change as $\Delta S = S_{grouped} - S_{ungrouped}$, where $S_{ungrouped}$ is the SIMBENCH score for simulating the general population and $S_{grouped}$ is the score when simulating a specific demographic group on the same question. A negative $\Delta S$ indicates that the model's simulation ability decreases relative to the uniform baseline when it is asked to simulate specific demographic groups.

Importantly, for SimBenchGrouped, we specifically selected questions where human response distributions showed the highest variance across demographic groups (see §2.4). The observed degradation in simulation performance therefore likely represents an upper bound on the challenges LLMs face when simulating specific demographic groups. Our results in Table 2 show that **LLMs struggle more with simulating specific demographic groups compared to general populations**. All evaluated models show negative mean $\Delta S$ values, with degradation ranging from -1.27 for DeepSeek-V3-0324 to -4.61 for Claude-3.7-Sonnet-4000. All degradations are statistically significant, with 95% confidence intervals excluding zero in every case.

Table 2: **Ungrouped vs. grouped simulation performance ($\Delta S$).** $\pm$ denotes 95% CI.

| Category | $\Delta S$ |
|---|---|
| **By Models** | |
| Claude-3.7-Sonnet | $-3.13_{\pm 0.66}$ |
| Claude-3.7-Sonnet-4000 | $-4.61_{\pm 0.65}$ |
| DeepSeek-R1 | $-3.79_{\pm 0.76}$ |
| DeepSeek-V3-0324 | $-1.27_{\pm 0.69}$ |
| GPT-4.1 | $-3.94_{\pm 0.76}$ |
| **By Demographics** | |
| Religiosity/Practice | $-9.91_{\pm 1.59}$ |
| Political Affil./Ideology | $-4.97_{\pm 1.28}$ |
| Religion (Affiliation) | $-4.83_{\pm 1.16}$ |
| Income/Social Standing | $-4.51_{\pm 1.23}$ |
| Domicile/Urbanicity | $-3.17_{\pm 0.82}$ |
| Employment Status | $-3.03_{\pm 0.82}$ |
| Education | $-2.55_{\pm 0.99}$ |
| Marital Status | $-1.80_{\pm 0.87}$ |
| Age | $-1.50_{\pm 0.85}$ |
| Gender | $-1.24_{\pm 0.72}$ |

The performance degradation varies substantially by demographic category. Models struggle most when simulating groups defined by religious attributes, with conditioning on "Religiosity/Practice" causing the largest decrease in simulation accuracy ($\Delta S = -9.91$), followed by "Political Affiliation/Ideology" ($\Delta S = -4.97$) and "Religion (Affiliation)" ($\Delta S = -4.83$). In contrast, models maintain relatively better performance when simulating groups defined by "Gender" ($\Delta S = -1.24$) and "Age" ($\Delta S = -1.50$).

While these findings may not fully generalize to cases where demographic differences are less pronounced, they highlight potential limitations in how current LLMs capture the nuanced response patterns of specific demographic groups. We argue that such challenging benchmarks are crucial for identifying areas where improvements are most needed, particularly for applications that aim to model the behaviors of specific subpopulations.

## 4.6 RQ6: SIMULATION ABILITY VS. GENERAL CAPABILITIES

Finally, we analyze the relationship between LLM simulation ability and more general model capabilities by correlating performance on SIMBENCH with popular LLM capability benchmarks. We collect performance data for eight models on five benchmarks representing distinct capabilities and calculate the Pearson correlation with their SIMBENCH scores (see Appendix J for implementation details and scatter plots).

We find that SIMBENCH performance correlates most strongly with benchmarks requiring knowledge-intensive reasoning, such as **MMLU-Pro (r = 0.94)** and **GPQA Diamond (r = 0.86)**. The correlation is weaker for general helpfulness (Chatbot Arena ELO, r = 0.71) and instruction following (IF-Eval, r = 0.79). Crucially, the correlation is substantially weaker for narrow, specialized skills like advanced mathematics (**OTIS AIME, r = 0.48**). We posit that accurately simulating human behavior is a complex capability rooted in broad, knowledge-intensive reasoning, which aligns with the diverse social and behavioral topics in SIMBENCH. The weaker correlations with Chatbot Arena and OTIS AIME suggest that neither general conversational ability nor narrow problem-solving skills are sufficient proxies for strong simulation performance.

## 5 RELATED WORK

**Human Behavior Simulation with LLMs** LLMs as human behavior simulators have attracted significant interdisciplinary attention. Researchers have evaluated their efficacy across political science (Argyle et al., 2023; Bisbee et al., 2024; Dominguez-Olmedo et al., 2024), psychology (Aher et al., 2023; Manning et al., 2024; Hewitt et al., 2024; Binz et al., 2025), economics (Horton, 2023; Aher et al., 2023), and computer science applications (Park et al., 2023; Hu & Collier, 2024; Dong et al., 2024; Hu & Collier, 2025). Evidence regarding LLMs' simulation fidelity remains mixed, with some studies reporting promising results (Argyle et al., 2023) while others identify critical limitations, including homogenized group representations (Cheng et al., 2023; Wang et al., 2025), a tendency toward hyper-rationality rather than human-like error (Liu et al., 2025a), and deterministic rather than distributional predictions (Park et al., 2024b).

Existing work has predominantly focused on individual-level simulation with minimal demographic conditioning, typically evaluating only one or two models in narrowly defined contexts. SIMBENCH addresses these limitations by providing a comprehensive benchmark for group-level simulation across diverse domains with systematic demographic conditioning and standardized metrics. SIMBENCH's distributional evaluation framework, based on Total Variation Distance, captures how accurately models represent the full spectrum of human response variation. This is an approach advocated for by researchers in both simulation (Anthis et al., 2025) and general LLM evaluation studies (Ying et al., 2025). For broader context on this emerging field, we refer readers to recent comprehensive surveys (Ma et al., 2024; Kozlowski & Evans, 2025; Olteanu et al., 2025; Anthis et al., 2025).

Appendix M continues our discussion of related work.

## 6 CONCLUSION AND FUTURE WORK

For LLM simulations to become reliable tools for the social and behavioral sciences, their fidelity to real human behavior must be measurable. However, prior evaluations have been fragmented, hindering systematic progress. To address this, we introduce SIMBENCH, the first large-scale, standardized benchmark for group-level human behavior simulation. By unifying 20 diverse datasets, SIMBENCH provides the necessary infrastructure to robustly evaluate and compare LLM simulators.

Using SIMBENCH, we provide the first systematic analysis of simulation ability across 45 LLMs. We show that even the best LLMs have limited simulation ability, that performance scales log-linearly with model size, and that there exists a fundamental tradeoff between standard alignment and simulating diverse human opinions. We further show that models struggle more when simulating specific demographic groups, and that strong simulation ability correlates with deep, knowledge-intensive reasoning. While significant progress is needed, SIMBENCH makes this progress measurable, providing an open foundation to accelerate the development of more faithful LLM simulators.

We hope that future research can build on this foundation by expanding beyond standardized formats to evaluate **interactive and open-ended behavioral simulations**. Finally, addressing the observed tradeoff between alignment and simulation fidelity requires developing **distribution-preserving alignment** techniques, alongside further investigation into the **causal mechanisms** linking other model capabilities to simulation performance.

## ACKNOWLEDGMENTS

This work was partially conducted while TH, JB, and PR were at the Data and Marketing Insights research unit of the Bocconi Institute for Data Science and Analysis. TH was supported by the Gates Cambridge Trust (grant OPP1144 from the Bill & Melinda Gates Foundation). JB acknowledges funding from the Swiss National Science Foundation (SNSF) through project P500-2_235328. DH was supported by the European Research Council (ERC) under the European Union's Horizon 2020 research and innovation program (grant agreement No. 949944, INTEGRATOR). PR was supported by a MUR FARE 2020 initiative under grant agreement Prot. R20YSMBZ8S (INDOMITA).

ETHICS STATEMENT

SIMBENCH's primary purpose is to benchmark LLM ability to simulate human behavior. While advancements in LLM simulation capabilities can support helpful applications such as pre-testing policies, these do not come without risks of misrepresentation and dual use.

RESPONSIBLE USE AND ACKNOWLEDGMENT OF LIMITATIONS

First and foremost, due to the observed limited simulation ability of state-of-the-art LLMs, we caution against relying on LLM-powered simulations of human behavior for tasks where downstream harm is possible. Even as models improve, substituting algorithmic approximations for authentic human participation carries the risk of disadvantaging under-represented / marginalized communities by removing their opportunities to directly shape decisions that affect them. Furthermore, while benchmarks like SIMBENCH help measure simulation capabilities, we must be careful not to mistake increasing benchmark performance for genuine understanding of complex human behavior.

DATA PROVENANCE AND TRANSFORMATIVE USE

The creation of SIMBENCH from 20 diverse sources was guided by a commitment to responsible data handling. Our curation process prioritized datasets with clear and permissive terms. As a result, 17 out of the 20 datasets are governed by explicit permissive licenses (e.g., Creative Commons, MIT). For the few remaining datasets that are publicly available for research without an explicit license, we apply a consistent framework built on the principle of transformative use.

1. **Transformative Use.** SIMBENCH does not contain or redistribute any raw, individual-level participant data. It is a new, derivative work consisting of aggregated, non-reversible group-level distributions. This process protects the privacy of the original human subjects.
2. **Multi-Level Licensing.** Our public release includes a detailed `LICENSE` file. **The SIMBENCH framework** (our code and pipeline) is permissively licensed (e.g., CC-BY-NC-SA 4.0). For each of the **20 constituent datasets**, the documentation explicitly lists the original source, its specific license or terms of use, and a clear statement clarifying its status as an aggregated, derivative work whose original terms should still be consulted.

SCOPE OF REPRESENTATION AND INTERSECTIONAL ANALYSIS

While SIMBENCH includes diverse demographic groups, it cannot adequately support simulations of intersectional identities due to sample size limitations. By conditioning on one demographic variable at a time, we cannot systematically assess how well models handle the rich overlap of identities (e.g., "older Latinx women," "young Black men"). This was a deliberate methodological choice to maintain the statistical integrity of the ground-truth distributions, as small intersectional group sizes make it difficult to combine multiple characteristics simultaneously due to increasing sampling noise in response distributions. Yet intersectional simulation is precisely where societal biases and model limitations often emerge, making this an important direction for future work. Additionally, the conditional prompting approach we use conceptualizes simplistic human populations and may thus fail to appropriately account for nuances of individual behavior.

CONCLUSION

Nevertheless, we believe SIMBENCH is an important step toward making LLM simulation progress measurable and raising awareness of state-of-the-art model blind spots. Together, we hope this will ultimately create accountability for models deployed in socially sensitive contexts.

REPRODUCIBILITY STATEMENT

SIMBENCH is released on HuggingFace. Our codebase is available on GitHub. We provide detailed descriptions of our experimental setup, including the exact prompts used for both base and instruction-tuned models, in Appendix D, and an empirical validation of our elicitation methodology in Appendix E.

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

# A    LIMITATIONS

**The Scope of Benchmarking Human Behavior**    The challenge of benchmarking human behavior lies in its vastness. No single study, nor dozens, could ever capture its full complexity. We operationalize this challenge by focusing on a core set of fundamental cognitive and social tasks widely studied in the behavioral sciences: decision-making, self-assessment, judgment, and problem-solving. This focus necessarily means that SIMBENCH does not capture the full complexity of human interaction, such as embodied or multi-turn social dynamics. As the first large-scale benchmark for group-level simulation, SIMBENCH provides the essential infrastructure to establish robust baselines, uncover fundamental properties like scaling trends, and map the frontier for future work on more complex, interactive simulations.

**Scope of Representativeness**    Although SIMBENCH spans 20 diverse datasets, the combined sample does (and can) not fully represent any single population in its full complexity. Many geographic regions are still underrepresented or entirely absent, potentially limiting generalizability to populations with different cultural backgrounds and preferences. Even within countries, demographic representativeness may vary, as only a subset of our 20 datasets are based on nationally representative sampling techniques. Each dataset carries its own statistical uncertainty. Opt-in samples and crowdsourced data (e.g., from Amazon Mechanical Turk) may have larger margins of error than nationally representative surveys, potentially affecting the benchmark's precision for certain questions. We view these limitations as opportunities for collaborative extension of SIMBENCH to improve global coverage and representativeness over time.

**Temporal Dimensions**    The current version of SIMBENCH utilizes static datasets that capture human behavior at specific points in time. This approach allows for systematic evaluation across domains but cannot yet assess how well LLMs simulate evolving preferences, opinion shifts, or behavioral adaptation – all fundamental aspects of human behavior. Future iterations of SIMBENCH could incorporate longitudinal data to address these dynamic aspects of human behavior and expand the benchmark's evaluative capacity.

**Task Format Considerations**    SIMBENCH currently focuses on multiple-choice, single-answer, single-turn questions and interactions. This standardized format enables systematic comparison across diverse domains but necessarily excludes more complex behavioral simulations including multi-step decision processes and interactive social dynamics. We see this as a pragmatic starting point that establishes foundational evaluation capabilities while inviting future extensions to capture more nuanced aspects of human behavior.

**Training Data Overlap**    A fundamental challenge for all LLM evaluation is the potential for training data contamination. While this cannot be definitively ruled out without full access to model training corpora, several aspects of our methodology and empirical findings suggest this risk is substantially mitigated for SIMBENCH:

1) Much of our source data is not easily ingested by standard web scrapes. Virtually all of our source data are primarily distributed as structured data files in specialized formats like R or SAS in academic archives. This makes data contamination much less likely than for generic web text, as these files cannot be meaningfully read or interpreted as plain text by standard scraping tools.

2) SimBench's core task is not to recall a fact but to predict a response distribution for a specific demographic subgroup (e.g., "women in Slovakia"). Even if a model's training data included thousands of individual survey responses, it would still need to learn, without supervision, how to aggregate these individual points into a coherent distribution for an arbitrary, specified subgroup. This is a sophisticated, zero-shot social reasoning skill that is unlikely to emerge from simply seeing the raw data.

3) On datasets that are most likely to appear in training data (e.g., US-centric OpinionQA), even the best-performing models achieve an S-score of only 60, far from the 100-point maximum. If models had memorized this benchmark, we would expect scores far closer to perfect. This clear performance ceiling demonstrates that our benchmark is testing a genuine capability rather than memorization.

4) The consistent scaling patterns we observe across diverse datasets suggest genuine simulation capabilities rather than artifacts of training data overlap.

Nevertheless, we acknowledge that data contamination remains a fundamental challenge in LLM evaluation, and future work should develop more robust methods to detect and quantify its impact. We include this consideration for completeness while believing it unlikely to significantly impact our current findings.

## B  DATASET CURATION AND EXCLUSION RATIONALE

As described in §2.1, our dataset curation process involved a systematic review of numerous prominent datasets in the social and behavioral sciences. While our search was extensive, many promising candidates were ultimately excluded for failing to meet our strict inclusion criteria for a redistributable, multiple-choice benchmark.

Table 3 provides an illustrative list of well-known datasets that were considered during this review but not included in the final SIMBENCH collection. This list is not exhaustive but serves to highlight the common methodological and logistical challenges that arise when creating a benchmark from existing scientific data, such as restrictive licensing, complex experimental designs, and non-standard response formats. By documenting these exclusion rationales, we aim to provide transparency into our curation process and offer a resource for future benchmark creators.

Table 3: Examples of Datasets Considered and Excluded from SIMBENCH.

| Dataset | Exclusion Reason |
|---|---|
| Understanding America Study | *Licensing Restrictions* |
| ManyLabs & ManyLabs 2 | *Complex treatment condition* |
| General Social Survey | *Licensing Restrictions* |
| Demographic and Health Surveys | *Licensing Restrictions* |
| Global Preferences Survey | *Licensing Restrictions* |
| World Risk Poll | *No raw question wording available* |
| Yourmorals.org | *Licensing Restrictions* |
| Asian Barometer | *Licensing Restrictions* |
| The Glasgow Norms | *Licensing Restrictions* |
| MovieLens | *Licensing Restrictions* |
| Health Information National Trends Survey | *Licensing Restrictions* |
| BBC Big Personality Test | *Licensing Restrictions* |
| Time-sharing Experiments for the Social Sciences | *Often complex treatment conditions* |
| Project Implicit | *Hard to model reaction time in LLMs* |
| Children's Worlds Survey | *Licensing Restrictions* |
| Monitoring the Future Survey | *Licensing Restrictions* |
| TIMSS | *Individual test items are not accessible* |
| UMD-OurDataHelps | *Free text response* |
| MobLab dataset | *Too few questions; lack detailed demographics data* |

## C  SIMBENCHPOP AND SIMBENCHGROUPED SAMPLING DETAILS

We curated data at two levels of grouping granularity, corresponding to our two main benchmark splits: **SimBenchPop** and **SimBenchGrouped**.

**SimBenchPop** measures LLMs' ability to simulate responses of broad, diverse human populations. We include all questions from all 20 datasets in SimBench, combining each question with its dataset-specific default grouping prompt (e.g., "You are an Amazon Mechanical Turk worker based in the United States"). We sample up to 500 questions per dataset to ensure representativeness while

keeping the benchmark manageable. For each test case, we aggregate individual responses across all participants in the dataset to create population-level response distributions. This approach creates a benchmark that represents population-level responses across diverse domains while maintaining a reasonable size of 7,167 test cases.

For **SimBenchGrouped**, we focus only on five large-scale survey datasets with rich demographic information and sufficient sample sizes: OpinionQA, ESS, Afrobarometer, ISSP, and Latinobarómetro. Our sampling approach prioritizes questions showing meaningful demographic variation. For each dataset, we identify available grouping variables (e.g., age, gender, country) with sufficient group sizes to form meaningful response distributions. We calculate the variance of responses across demographic groups for each question and rank questions by their variance scores, prioritizing those showing the strongest demographic differences. We select questions that exhibit significant variation across demographic groups to ensure the benchmark captures meaningful differences in responses. For each selected question, we create multiple test cases by pairing it with different values of the grouping variables (e.g., age = "18-29", age = "30-49"). This process results in 6,343 test cases that specifically measure LLMs' ability to simulate responses from narrower participant groups based on specified demographic characteristics. Table 4 provides a summary of the sampling process across all datasets, showing the minimum group size thresholds and the number of test cases in each benchmark split.

Table 4: Dataset Sampling Summary; NaN refers to a dataset that is only available in aggregated form and no grouping size is known.

| Dataset | Min. Group | SimBench | SimBenchPop | SimBenchGrouped |
|---|---|---|---|---|
| WisdomOfCrowds | 100 | 1,604 | 114 | – |
| Jester | 100 | 136 | 136 | – |
| Choices13k | NaN | 14,568 | 500 | – |
| OpinionQA | 300 | 1,074,392 | 500 | 984 |
| MoralMachineClassic | 100 | 3,441 | 15 | – |
| MoralMachine | 100 | 20,771 | 500 | – |
| ChaosNLI | 100 | 4,645 | 500 | – |
| ESS | 300 | 2,783,780 | 500 | 1,643 |
| Afrobarometer | 300 | 517,453 | 500 | 1,531 |
| OSPsychBig5 | 300 | 1,950 | 250 | – |
| OSPsychMACH | 300 | 3,682,700 | 100 | – |
| OSPsychMGKT | 300 | 20,610 | 500 | – |
| OSPsychRWAS | 300 | 975,585 | 22 | – |
| ISSP | 300 | 594,336 | 500 | 940 |
| Latinobarómetro | 300 | 80,684 | 500 | 1,245 |
| GlobalOpinionQA | NaN | 46,329 | 500 | – |
| DICES | 10 | 918,064 | 500 | – |
| NumberGame | 10 | 15,984 | 500 | – |
| ConspiracyCorr | 300 | 968 | 45 | – |
| TISP | 300 | 172,271 | 485 | – |
| **Total** | | **10,930,271** | **7,167** | **6,343** |

## D   IMPLEMENTATION DETAILS

For base models, we use HuggingFace Transformers (Wolf et al., 2020) to run inference on a single NVIDIA RTX A6000 Ada GPU. We structure prompts so that the next token corresponds to the model's answer choices. For models smaller than 70B parameters, we use 8-bit quantization implemented in bitsandbytes (Dettmers et al., 2022), while 70B models use 4-bit quantization.

For instruction-tuned models, we use API calls. OpenAI models are accessed directly through their API, while other models are accessed via OpenRouter. We request verbalized probability outputs in JSON format with temperature initially set to 0. If parsing fails, we increase temperature to 1 and

retry up to 5 times. All models successfully produced valid JSON under these conditions. When probability outputs do not sum to 1, we apply normalization.

Our evaluation includes a diverse set of models: Qwen 2.5 (Yang et al., 2024) (0.5B-72B), Gemma 3 (Team et al., 2025) PT and IT (4B-27B), o4-mini (OpenAI, 2025b), Claude 3.7 Sonnet (Anthropic, 2025), DeepSeek R1 (Guo et al., 2025), DeepSeek-V3-0324 DeepSeek-AI (2024), GPT-4.1 OpenAI (2025a), and Llama-3.1-Instruct (8B-405B) (Meta AI, 2024).

To ensure the validity of our results, we perform two checks: 1) We verify that base models assign the vast majority of probability mass to the provided answer options. Even for small models like Qwen2.5-0.5B, the sum of probabilities across answer tokens is as high as 0.98, confirming that models rarely predict tokens outside the designated answer space. 2) We also evaluate the effect of quantization on model performance using a subset of SimBench. As shown in Table 5, performance remains consistent across quantization levels, with minimal variation in total variation scores even for quantization-sensitive models like Llama-3.1.

We detail below the prompts used in our experimental conditions for token probability and verbalized distribution prediction.

The following system prompt was consistent across all experimental conditions:

> You are a group of individuals with these shared characteristics:
> {default system prompt}{grouping system prompt (if any)}

For token probability prediction, we adapted the prompt structure from Nori et al. (2023):

> **Question**: {question}
> Do not provide any explanation, only answer with one of the following options: {answer options}.
> **Answer**: (

Prompt for eliciting verbalized probability prediction:

> **Question**: {question}
> Estimate what percentage of your group would choose each option. Follow these rules:
> 1. Use whole numbers from 0 to 100
> 2. Ensure the percentages sum to exactly 100
> 3. Only include the numbers (no % symbols)
> 4. Use this exact valid JSON format: {answer options} and do NOT include anything else.
> 5. Only output your final answer and nothing else. No explanations or intermediate steps are
> ↪    needed.
> Replace X with your estimated percentages for each option.
> '**Answer**:

Prompt for zero-shot CoT:

> **Question**: {question}
> Estimate what percentage of your group would choose each option.
> Think step by step about how people with your shared characteristics would reason about this
> ↪    question.
> Consider different perspectives within your group and what factors would influence their choices.
>
> Please provide your reasoning first, then give your final answer in JSON format.
> Follow these rules for your final answer:
> 1. Use whole numbers from 0 to 100
> 2. Ensure the percentages sum to exactly 100
> 3. Only include the numbers (no % symbols)
> 4. Use this exact valid JSON format: {json_format_str}
> 5. Replace X with your estimated percentages for each option.
> '**Answer**:

# E    VALIDATION OF ELICITATION METHOD

A key methodological choice in SIMBENCH is how to elicit probability distributions from LLMs. For base models, we use direct token probabilities from the first token of the response. For instruction-tuned models, however, two primary methods exist: direct token probabilities and requesting a "verbalized" distribution (e.g., a JSON object with percentages). To validate our choice of using verbalized distributions for instruction-tuned models, as recommended by recent work (Tian et al., 2023; Meister et al., 2025), we conducted a direct comparison.

Figure 5 compares the SIMBENCH scores for several instruction-tuned models using both methods. The results are unequivocal: using verbalized distributions (teal dots) dramatically and consistently outperforms direct token probabilities (orange dots) for every instruction-tuned model tested. In many cases, using token probabilities results in scores far below zero, indicating that the model's raw logits are poorly calibrated for this task after instruction tuning. In contrast, base models (black bars) perform reasonably well with token probabilities, as they are not subject to the same post-training shifts.

This analysis provides strong empirical support for our methodological decision to use token probabilities for base models and verbalized distributions for instruction-tuned models, ensuring that we are evaluating each model class using the most effective and well-calibrated elicitation technique.

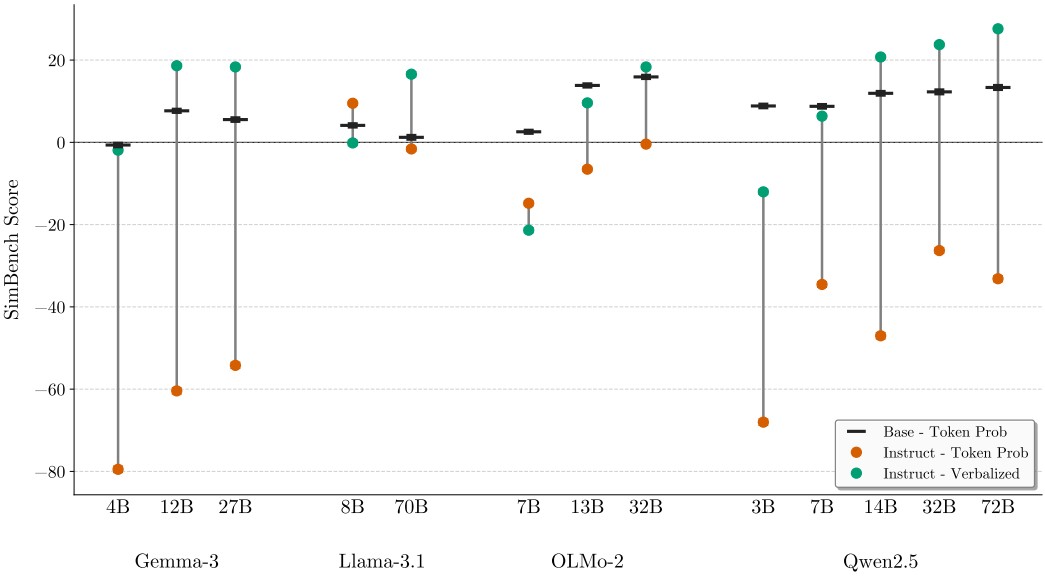

Figure 5: **Verbalized distributions are superior for instruction-tuned models.** This plot compares SIMBENCH scores for base models (using token probabilities) against their instruction-tuned counterparts using either token probabilities (orange) or verbalized distributions (teal). The large vertical gap for all instruction-tuned models demonstrates the significant performance gain from using verbalized distributions, validating our choice of elicitation method.

Table 5: Total Variation for different models at various quantization levels. Lower values indicate better performance.

| Model | 4-bit | 8-bit | 16-bit | 32-bit |
|---|---|---|---|---|
| Llama-3.1-8B-Instruct | 0.272 | 0.266 | 0.262 | 0.262 |
| Qwen2.5-7B | 0.307 | 0.307 | 0.306 | 0.307 |

## F    METRIC ROBUSTNESS CHECK

TVD ranges from 0 (perfect match) to 1 (complete disagreement), with lower values indicating better simulation fidelity. TVD provides an interpretable measure of how closely model predictions align with actual human response distributions. TVD is particularly well-suited for simulation evaluation compared to alternatives like KL divergence or Jensen-Shannon divergence (JSD). Unlike KL divergence, TVD remains well-defined even when the model assigns zero probability to responses that humans give, avoiding the infinite penalties that KL would impose in such cases. Additionally, TVD is symmetric and bounded, making it more interpretable across different datasets and response distributions than KL divergence. While JSD offers similar advantages in terms of symmetry and boundedness, TVD provides a more direct and intuitive interpretation of the maximum possible error in probability estimates. This property is especially valuable when evaluating how accurately models simulate the distribution of human responses rather than just matching the most likely response. For further discussion on TVD as an evaluation metric, see also Meister et al. (2025). We show the results of Table 1 in terms of raw TVD values in Table 10.

To ensure our findings are robust across different metrics, we complement TVD with two alternative metrics: Jensen-Shannon Divergence (JSD) and Spearman's Rank Correlation (RC). Table 6 presents these metrics for a subset of evaluated models. The strong Pearson correlation between TVD and JSD ($r = 0.92$) indicates these metrics provide consistent model rankings. The moderate negative correlation ($r = -0.57$) between TVD and RC is expected, as lower distances correspond to higher correlations. This multi-metric evaluation confirms that our model comparisons remain consistent across different statistical measures.

We chose the uniform distribution as the primary baseline U because it represents a state of maximum uncertainty, or the "zero-knowledge" guess. This provides the most conservative and universally applicable baseline across questions with varying numbers of choices. While other baselines, such as a majority-class baseline, could be considered, they would incorporate some knowledge about the distribution, making the $S$=0 point less interpretable as a true "no-skill" score.

Table 6: Comparison of models on three metrics: Total Variation Distance (TVD), Jensen-Shannon Divergence (JSD), and Spearman Rank Correlation (RC). Lower values are better for TVD and JSD; higher is better for RC.

| Model | Total Variation | JS Divergence | Rank Correlation |
|---|---|---|---|
| Claude-3.7-Sonnet | 0.191 | 0.057 | 0.673 |
| Claude-3.7-Sonnet-4000 | 0.195 | 0.060 | 0.648 |
| DeepSeek-R1 | 0.211 | 0.069 | 0.623 |
| DeepSeek-V3-0324 | 0.216 | 0.069 | 0.620 |
| GPT-4.1 | 0.209 | 0.070 | 0.646 |
| Llama-3.1-405B-Instruct | 0.231 | 0.085 | 0.593 |
| o4-mini-high | 0.225 | 0.079 | 0.621 |
| o4-mini-low | 0.230 | 0.082 | 0.609 |

## G    FULL SIMBENCH RESULTS (RQ1)

We show the SIMBENCH scores for all 45 models we evaluate in Table 7. We show the scaling law plots for all models in Figure 6.

## H    STATISTICAL ANALYSIS OF MODEL PERFORMANCE

To ensure the robustness of our findings, we conducted a comprehensive statistical analysis of model performance on SIMBENCH. This includes confidence intervals for overall scores, pairwise significance tests between models, and within-family analyses to validate our scaling law observations.

Table 7: **Overall simulation ability** as measured by SIMBENCH score $S$ averaged across the two main splits of SIMBENCH. Reasoning models are highlighted in *italics*. Models are sorted by score. Models below the dotted line perform worse than a uniform baseline.

| Model | Type | Release | $S$ ($\uparrow$) |
|---|---|---|---|
| Claude-3.7-Sonnet | Instr. | Closed | 40.80 |
| *Claude-3.7-Sonnet-4000* | Instr. | Closed | 39.46 |
| GPT-4.1 | Instr. | Closed | 34.55 |
| *DeepSeek-R1* | Instr. | Open | 34.52 |
| DeepSeek-V3-0324 | Instr. | Open | 32.89 |
| *o4-mini-high* | Instr. | Closed | 28.99 |
| Llama-3.1-405B-Instruct | Instr. | Open | 28.40 |
| *o4-mini-low* | Instr. | Closed | 27.77 |
| Qwen2.5-72B-Instruct | Instr. | Open | 27.61 |
| Qwen2.5-32B-Instruct | Instr. | Open | 23.76 |
| Qwen2.5-14B-Instruct | Instr. | Open | 20.75 |
| OLMo-2-0325-32B-DPO | Instr. | Open | 19.80 |
| Gemma-3-12B-IT | Instr. | Open | 18.62 |
| Gemma-3-27B-IT | Instr. | Open | 18.33 |
| OLMo-2-0325-32B-Instruct | Instr. | Open | 18.32 |
| Llama-3.1-70B-Instruct | Instr. | Open | 16.56 |
| OLMo-2-0325-32B | Base | Open | 15.90 |
| Llama-3.1-Minitaur-8B | Base | Open | 14.50 |
| OLMo-2-1124-13B | Base | Open | 13.83 |
| Qwen2.5-72B | Base | Open | 13.34 |
| Qwen2.5-32B | Base | Open | 12.27 |
| Qwen2.5-14B | Base | Open | 11.92 |
| OLMo-2-0325-32B-SFT | Instr. | Open | 11.28 |
| OLMo-2-1124-13B-Instruct | Instr. | Open | 9.59 |
| OLMo-2-1124-13B-DPO | Instr. | Open | 9.42 |
| Qwen2.5-3B | Base | Open | 8.84 |
| Qwen2.5-7B | Base | Open | 8.75 |
| Llama-3.1-Centaur-70B | Base | Open | 8.54 |
| Gemma-3-12B-PT | Base | Open | 7.66 |
| Qwen2.5-7B-Instruct | Instr. | Open | 6.36 |
| Gemma-3-27B-PT | Base | Open | 5.53 |
| Qwen2.5-1.5B | Base | Open | 5.34 |
| Llama-3.1-8B | Base | Open | 4.12 |
| OLMo-2-1124-7B | Base | Open | 2.56 |
| Llama-3.1-70B | Base | Open | 1.21 |
| Llama-3.1-8B-Instruct | Instr. | Open | -0.15 |
| Gemma-3-4B-PT | Base | Open | -0.65 |
| Gemma-3-4B-IT | Instr. | Open | -1.91 |
| Qwen2.5-0.5B | Base | Open | -3.00 |
| OLMo-2-1124-7B-SFT | Instr. | Open | -11.36 |
| Qwen2.5-3B-Instruct | Instr. | Open | -12.04 |
| Gemma-3-1B-PT | Base | Open | -16.17 |
| OLMo-2-1124-7B-DPO | Instr. | Open | -19.62 |
| OLMo-2-1124-13B-SFT | Instr. | Open | -20.54 |
| OLMo-2-1124-7B-Instruct | Instr. | Open | -21.36 |

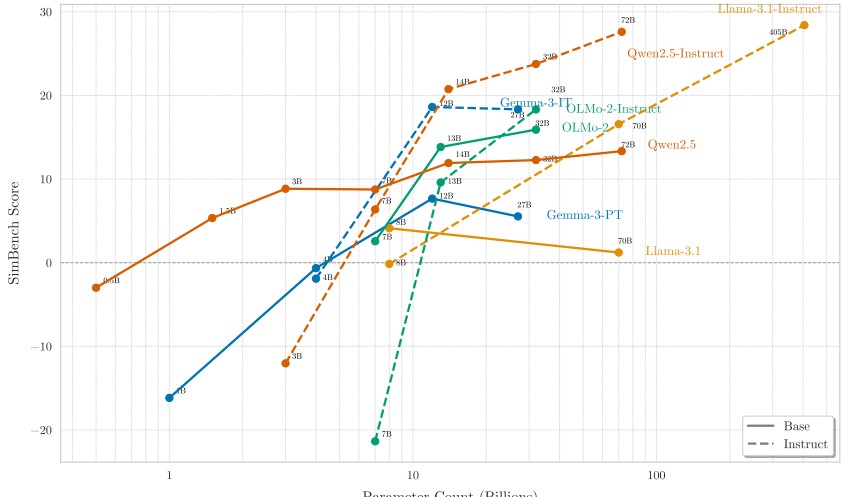

Figure 6: **Model parameter count vs. simulation ability**. We measure model size by parameter count and simulation ability by SIMBENCH score $S$ averaged across the two main splits of SIMBENCH.

## H.1 OVERALL MODEL RANKING AND SIGNIFICANCE

Table 8 expands on the results from Table 1 in the main paper. We report the 95% confidence interval (CI) for each model's mean SIMBENCH score. We also perform independent two-sample t-tests to determine if the performance difference between each model and the one ranked immediately below it is statistically significant. The results confirm a clear and statistically robust performance hierarchy among the evaluated models.

Table 8: Detailed statistical analysis of SIMBENCH scores for the top 8 models. Pairwise significance tests compare each model to the one ranked immediately below it. Symbols indicate significance: $^*p < 0.05$, $^{**}p < 0.01$, $^{***}p < 0.001$.

| Rank | Model | Score (S) $\pm$ SE | 95% CI |
|------|-------|--------------------|--------|
| 1 | Claude-3.7-Sonnet | $40.80(43)^*$ | [39.96, 41.64] |
| 2 | Claude-3.7-Sonnet-4000 | $39.46(42)^{***}$ | [38.63, 40.29] |
| 3 | GPT-4.1 | $34.55(48)$ | [33.62, 35.50] |
| 4 | DeepSeek-R1 | $34.52(43)^{**}$ | [33.68, 35.37] |
| 5 | DeepSeek-V3-0324 | $32.89(42)^{***}$ | [32.07, 33.72] |
| 6 | o4-mini-high | $28.99(52)$ | [27.97, 30.02] |
| 7 | Llama-3.1-405B-Instruct | $28.40(49)$ | [27.46, 29.36] |
| 8 | o4-mini-low | $27.77(51)^{***}$ | [26.77, 28.78] |

## H.2 WITHIN-FAMILY PERFORMANCE ANALYSIS

To investigate the impact of model size (RQ2), we conducted one-way ANOVA followed by post-hoc Tukey HSD tests for pairwise comparisons within model families. The results, presented in Table 9, show that for most families, increases in model size lead to statistically significant improvements in simulation ability, supporting the scaling trends observed in Figure 2.

Table 9: Within-family pairwise performance comparisons. All differences are statistically significant at $p < 0.001$ unless otherwise noted (ns).

| Model Family | Comparison | Result |
|---|---|---|
| **Llama-3.1-Instruct** (ANOVA: F=518.8, p<0.001) | 405B vs 70B | $\Delta S = 11.84 \ (p < 0.001)$ |
| | 405B vs 8B | $\Delta S = 28.55 \ (p < 0.001)$ |
| | 70B vs 8B | $\Delta S = 16.71 \ (p < 0.001)$ |
| **Gemma-3-IT** (ANOVA: F=710.1, p<0.001) | 27B vs 12B | $\Delta S = 0.29 \ (\text{ns}, \ p = 0.673)$ |
| | 12B vs 4B | $\Delta S = 20.54 \ (p < 0.001)$ |
| | 27B vs 4B | $\Delta S = 20.24 \ (p < 0.001)$ |
| **Qwen2.5-Instruct** (ANOVA: F=518.8, p<0.001) | 72B vs 32B | $\Delta S = 3.85 \ (p < 0.001)$ |
| | 72B vs 14B | $\Delta S = 6.86 \ (p < 0.001)$ |
| | 72B vs 7B | $\Delta S = 21.24 \ (p < 0.001)$ |
| | 32B vs 14B | $\Delta S = 3.01 \ (p < 0.001)$ |
| | 32B vs 7B | $\Delta S = 17.39 \ (p < 0.001)$ |
| | 14B vs 7B | $\Delta S = 14.38 \ (p < 0.001)$ |

# I ANALYSIS OF THE ALIGNMENT-SIMULATION TRADEOFF

## I.1 OBSERVATIONAL ANALYSIS

As a complement to the analysis in §4.4, this section provides a direct observational comparison of how base and instruction-tuned models perform as a function of human response plurality. We operationalize response plurality as the normalized entropy of the human response distribution, where high entropy indicates widespread disagreement while a low entropy indicates strong consensus. Simulation fidelity is measured by Total Variation Distance (TVD), where lower values indicate better performance. Figure 7 visualizes this relationship for all question-model pairs in SimBenchPop, separated by model type. The plots reveal a clear and divergent behavioral pattern:

- **Base models** exhibit a negative correlation between entropy and TVD. This demonstrates an observable property of this model class: they are generally more accurate (lower TVD) on high-entropy questions where human opinions are diverse.

- **Instruction-tuned models** exhibit a positive correlation. This demonstrates the opposite property: they are generally more accurate on low-entropy questions where humans have reached a consensus.

This direct visualization establishes a core empirical finding: the two model classes have fundamentally different strengths when simulating human responses across the spectrum of opinion plurality. The analysis in the main paper (§4.4) builds upon this observation to more formally quantify the *effect* of post-training that drives this divergence.

## I.2 REGRESSION ANALYSIS

To formally test the relationship between human response entropy and simulation performance across different model types, we fit an Ordinary Least Squares (OLS) regression model predicting Total Variation (TV) distance at the individual question-model level. The model specification was as follows:

$$\text{Total\_Variation} \sim C(\text{dataset\_name}) + C(\text{model}) + C(\text{instruct\_flag}) : \text{Human\_Normalized\_Entropy} \tag{3}$$

Here, *Total_Variation* is the dependent variable. $C(\text{dataset\_name})$ and $C(\text{model})$ represent fixed effects for each dataset and model, respectively, controlling for baseline differences in difficulty and capability. The crucial term is the interaction $C(\text{instruct\_flag}) : \text{Human\_Normalized\_Entropy}$, where *instruct_flag* is a binary indicator for instruction-tuned models (0 for base, 1 for instruction-tuned).

The key results from Table 11 are the coefficients for the interaction terms:

Table 10: TVD for each model in SimBenchPop and SimBenchGrouped. Lower values indicate better performance. PT and IT refer to pretrained and instruction-tuned versions, respectively.

| Model | SimBenchPop | SimBenchGrouped | Average |
|---|---|---|---|
| *Baselines* | | | |
| Uniform baseline | 0.335 | 0.362 | 0.348 |
| *Models* | | | |
| Claude-3.7-Sonnet | 0.197 | 0.183 | 0.191 |
| Claude-3.7-Sonnet-4000 | 0.201 | 0.189 | 0.195 |
| GPT-4.1 | 0.212 | 0.206 | 0.209 |
| DeepSeek-R1 | 0.211 | 0.212 | 0.211 |
| DeepSeek-V3-0324 | 0.215 | 0.218 | 0.216 |
| o4-mini-high | 0.235 | 0.214 | 0.225 |
| o4-mini-low | 0.234 | 0.226 | 0.230 |
| Llama-3.1-405B-Instruct | 0.237 | 0.225 | 0.231 |
| Qwen2.5-72B-Instruct | 0.229 | 0.246 | 0.237 |
| Qwen2.5-32B-Instruct | 0.242 | 0.258 | 0.250 |
| OLMo-2-0325-32B-DPO | 0.258 | 0.258 | 0.258 |
| Qwen2.5-14B-Instruct | 0.247 | 0.270 | 0.258 |
| OLMo-2-0325-32B-Instruct | 0.263 | 0.260 | 0.261 |
| Llama-3.1-70B-Instruct | 0.277 | 0.247 | 0.263 |
| Gemma-3-12B-IT | 0.262 | 0.274 | 0.267 |
| Gemma-3-27B-IT | 0.270 | 0.273 | 0.272 |
| OLMo-2-0325-32B | 0.271 | 0.297 | 0.283 |
| OLMo-2-0325-32B-SFT | 0.298 | 0.265 | 0.283 |
| Qwen2.5-72B | 0.268 | 0.300 | 0.283 |
| Qwen2.5-32B | 0.273 | 0.308 | 0.290 |
| Llama-3.1-Minitaur-8B | 0.288 | 0.296 | 0.292 |
| OLMo-2-1124-13B | 0.284 | 0.302 | 0.293 |
| Qwen2.5-14B | 0.285 | 0.314 | 0.298 |
| OLMo-2-1124-13B-DPO | 0.293 | 0.306 | 0.299 |
| OLMo-2-1124-13B-Instruct | 0.295 | 0.304 | 0.299 |
| Qwen2.5-7B | 0.290 | 0.326 | 0.307 |
| Llama-3.1-Centaur-70B | 0.309 | 0.313 | 0.311 |
| Qwen2.5-7B-Instruct | 0.292 | 0.332 | 0.311 |
| Qwen2.5-3B | 0.300 | 0.327 | 0.313 |
| Gemma-3-12B-PT | 0.310 | 0.317 | 0.314 |
| Gemma-3-27B-PT | 0.309 | 0.325 | 0.317 |
| Llama-3.1-8B-Instruct | 0.321 | 0.318 | 0.320 |
| Qwen2.5-1.5B | 0.321 | 0.324 | 0.322 |
| Llama-3.1-8B | 0.326 | 0.323 | 0.324 |
| Llama-3.1-70B | 0.331 | 0.324 | 0.328 |
| OLMo-2-1124-7B | 0.324 | 0.349 | 0.336 |
| Gemma-3-4B-PT | 0.334 | 0.341 | 0.337 |
| Gemma-3-4B-IT | 0.337 | 0.341 | 0.339 |
| Qwen2.5-0.5B | 0.337 | 0.364 | 0.349 |
| OLMo-2-1124-7B-SFT | 0.393 | 0.355 | 0.375 |
| Qwen2.5-3B-Instruct | 0.397 | 0.363 | 0.381 |
| Gemma-3-1B-PT | 0.382 | 0.414 | 0.397 |
| OLMo-2-1124-7B-DPO | 0.413 | 0.382 | 0.399 |
| OLMo-2-1124-7B-Instruct | 0.420 | 0.386 | 0.404 |
| OLMo-2-1124-13B-SFT | 0.416 | 0.414 | 0.415 |

Table 11: Results: Ordinary least squares

| Model: | OLS | Adj. R-squared: | 0.168 |
|---|---|---|---|
| Dependent Variable: | Total_Variation | AIC: | -163587.0704 |
| Date: | 2025-09-23 12:11 | BIC: | -163085.0895 |
| No. Observations: | 207837 | Log-Likelihood: | 81843. |
| Df Model: | 48 | F-statistic: | 875.3 |
| Df Residuals: | 207788 | Prob (F-statistic): | 0.00 |
| R-squared: | 0.168 | Scale: | 0.026644 |

| | Coef. | Std.Err. | t | P> \|t\| | [0.025 | 0.975] |
|---|---|---|---|---|---|---|
| Intercept | 0.1811 | 0.0024 | 75.6586 | 0.0000 | 0.1764 | 0.1858 |
| C(dataset_name)[T.ChaosNLI] | -0.0372 | 0.0019 | -19.2201 | 0.0000 | -0.0409 | -0.0334 |
| C(dataset_name)[T.Choices13k] | -0.0972 | 0.0019 | -49.9249 | 0.0000 | -0.1010 | -0.0934 |
| C(dataset_name)[T.ConspiracyCorr] | -0.0240 | 0.0047 | -5.0722 | 0.0000 | -0.0333 | -0.0147 |
| C(dataset_name)[T.DICES] | -0.0331 | 0.0021 | -15.8245 | 0.0000 | -0.0372 | -0.0290 |
| C(dataset_name)[T.ESS] | -0.0261 | 0.0019 | -13.5145 | 0.0000 | -0.0299 | -0.0223 |
| C(dataset_name)[T.GlobalOpinionQA] | -0.0406 | 0.0019 | -21.1621 | 0.0000 | -0.0444 | -0.0369 |
| C(dataset_name)[T.ISSP] | -0.0292 | 0.0019 | -15.1813 | 0.0000 | -0.0330 | -0.0255 |
| C(dataset_name)[T.Jester] | 0.1104 | 0.0029 | 37.4254 | 0.0000 | 0.1046 | 0.1162 |
| C(dataset_name)[T.Latinobarómetro] | -0.0368 | 0.0019 | -18.9530 | 0.0000 | -0.0406 | -0.0330 |
| C(dataset_name)[T.MoralMachine] | -0.0438 | 0.0019 | -22.6987 | 0.0000 | -0.0476 | -0.0401 |
| C(dataset_name)[T.MoralMachineClassic] | -0.1676 | 0.0079 | -21.0886 | 0.0000 | -0.1831 | -0.1520 |
| C(dataset_name)[T.NumberGame] | -0.0852 | 0.0019 | -44.4242 | 0.0000 | -0.0889 | -0.0814 |
| C(dataset_name)[T.OSPsychBig5] | -0.1218 | 0.0024 | -51.0313 | 0.0000 | -0.1265 | -0.1171 |
| C(dataset_name)[T.OSPsychMACH] | -0.0200 | 0.0033 | -5.9606 | 0.0000 | -0.0265 | -0.0134 |
| C(dataset_name)[T.OSPsychMGKT] | -0.1121 | 0.0019 | -57.9995 | 0.0000 | -0.1159 | -0.1083 |
| C(dataset_name)[T.OSPsychRWAS] | 0.0105 | 0.0066 | 1.5909 | 0.1116 | -0.0024 | 0.0235 |
| C(dataset_name)[T.OpinionQA] | -0.1080 | 0.0019 | -56.3479 | 0.0000 | -0.1118 | -0.1043 |
| C(dataset_name)[T.TISP] | -0.0429 | 0.0020 | -21.9962 | 0.0000 | -0.0467 | -0.0391 |
| C(dataset_name)[T.WisdomOfCrowds] | -0.0224 | 0.0032 | -7.0814 | 0.0000 | -0.0286 | -0.0162 |
| C(Model)[T.DeepSeek-R1] | 0.0114 | 0.0024 | 4.8093 | 0.0000 | 0.0067 | 0.0160 |
| C(Model)[T.DeepSeek-V3-0324] | 0.0158 | 0.0024 | 6.6887 | 0.0000 | 0.0112 | 0.0204 |
| C(Model)[T.GPT-4.1] | 0.0122 | 0.0024 | 5.1618 | 0.0000 | 0.0076 | 0.0168 |
| C(Model)[T.Gemma-3-12B-IT] | 0.0622 | 0.0024 | 26.3302 | 0.0000 | 0.0575 | 0.0668 |
| C(Model)[T.Gemma-3-12B-PT] | 0.3521 | 0.0031 | 115.0964 | 0.0000 | 0.3461 | 0.3581 |
| C(Model)[T.Gemma-3-27B-IT] | 0.0711 | 0.0024 | 30.1015 | 0.0000 | 0.0665 | 0.0757 |
| C(Model)[T.Gemma-3-27B-PT] | 0.3509 | 0.0031 | 114.6963 | 0.0000 | 0.3449 | 0.3569 |
| C(Model)[T.Llama-3.1-405B-Instruct] | 0.0373 | 0.0024 | 15.7766 | 0.0000 | 0.0326 | 0.0419 |
| C(Model)[T.Llama-3.1-70B] | 0.3730 | 0.0031 | 121.9329 | 0.0000 | 0.3670 | 0.3790 |
| C(Model)[T.Llama-3.1-70B-Instruct] | 0.0772 | 0.0024 | 32.7116 | 0.0000 | 0.0726 | 0.0819 |
| C(Model)[T.Llama-3.1-8B] | 0.3676 | 0.0031 | 120.1756 | 0.0000 | 0.3616 | 0.3736 |
| C(Model)[T.Llama-3.1-8B-Instruct] | 0.1212 | 0.0024 | 51.3289 | 0.0000 | 0.1166 | 0.1258 |
| C(Model)[T.OLMo-2-0325-32B] | 0.3125 | 0.0031 | 102.1507 | 0.0000 | 0.3065 | 0.3185 |
| C(Model)[T.OLMo-2-0325-32B-Instruct] | 0.0636 | 0.0024 | 26.9284 | 0.0000 | 0.0590 | 0.0682 |
| C(Model)[T.OLMo-2-1124-13B] | 0.3261 | 0.0031 | 106.5964 | 0.0000 | 0.3201 | 0.3321 |
| C(Model)[T.OLMo-2-1124-13B-Instruct] | 0.0953 | 0.0024 | 40.3576 | 0.0000 | 0.0907 | 0.0999 |
| C(Model)[T.Qwen2.5-1.5B] | 0.3624 | 0.0031 | 118.4639 | 0.0000 | 0.3564 | 0.3684 |
| C(Model)[T.Qwen2.5-14B] | 0.3264 | 0.0031 | 106.6913 | 0.0000 | 0.3204 | 0.3324 |
| C(Model)[T.Qwen2.5-14B-Instruct] | 0.0481 | 0.0024 | 20.3827 | 0.0000 | 0.0435 | 0.0528 |
| C(Model)[T.Qwen2.5-32B] | 0.3152 | 0.0031 | 103.0509 | 0.0000 | 0.3092 | 0.3212 |
| C(Model)[T.Qwen2.5-32B-Instruct] | 0.0426 | 0.0024 | 18.0362 | 0.0000 | 0.0380 | 0.0472 |
| C(Model)[T.Qwen2.5-3B] | 0.3422 | 0.0031 | 111.8592 | 0.0000 | 0.3362 | 0.3482 |
| C(Model)[T.Qwen2.5-72B] | 0.3103 | 0.0031 | 101.4261 | 0.0000 | 0.3043 | 0.3163 |
| C(Model)[T.Qwen2.5-72B-Instruct] | 0.0292 | 0.0024 | 12.3690 | 0.0000 | 0.0246 | 0.0338 |
| C(Model)[T.Qwen2.5-7B] | 0.3314 | 0.0031 | 108.3263 | 0.0000 | 0.3254 | 0.3374 |
| C(Model)[T.o4-mini-high] | 0.0354 | 0.0024 | 15.0086 | 0.0000 | 0.0308 | 0.0401 |
| C(Model)[T.o4-mini-low] | 0.0344 | 0.0024 | 14.5682 | 0.0000 | 0.0298 | 0.0390 |
| C(instruct_flag)[base]:Human_Normalized_Entropy | -0.2451 | 0.0024 | -100.0564 | 0.0000 | -0.2499 | -0.2403 |
| C(instruct_flag)[instruct]:Human_Normalized_Entropy | 0.0997 | 0.0022 | 46.1412 | 0.0000 | 0.0955 | 0.1039 |

| Omnibus: | 32553.497 | Durbin-Watson: | 1.732 |
|---|---|---|---|
| Prob(Omnibus): | 0.000 | Jarque-Bera (JB): | 60320.484 |
| Skew: | 0.995 | Prob(JB): | 0.000 |
| Kurtosis: | 4.733 | Condition No.: | 30 |

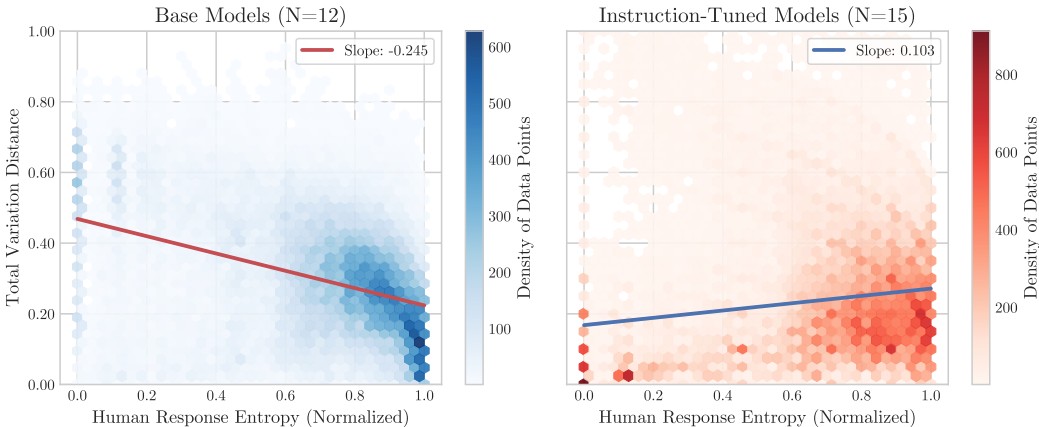

Figure 7: **Response plurality vs. simulation fidelity** for base and instruction-tuned models on all questions in SimBenchPop. We measure response plurality by normalized entropy of the human response distribution and simulation fidelity by total variation distance at the question level.

- For base models: The coefficient on the interaction between base models and Human Normalized Entropy is $-0.2451$ ($p < 0.001$), indicating that for every one-unit increase in normalized entropy, the TVD decreases by approximately 0.25 units. This means that base models perform *better* (lower TVD) when simulating human populations with more diverse opinions.

- For instruction-tuned models: The coefficient on the interaction between instruction-tuned models and Human Normalized Entropy is $+0.0997$ ($p < 0.001$), indicating that for every one-unit increase in normalized entropy, the TVD increases by approximately 0.11 units. This means that instruction-tuned models perform *worse* (higher TVD) when simulating human populations with more diverse opinions.

These coefficients are both highly statistically significant ($p < 0.001$) and represent substantial effect sizes given that TVD ranges from 0 to 1. The model as a whole explains approximately 17% of the variance in TVD ($R^2 = 0.168$), which is substantial for a dataset of this size and complexity. The opposite signs of these coefficients provide strong evidence for our hypothesis that base models and instruction-tuned models respond differently to the challenge of simulating populations with diverse opinions. This pattern holds even after controlling for the specific datasets and models involved, suggesting it represents a general property of the two model classes rather than an artifact of particular model or evaluation datasets.

### I.3 CAUSAL MEDIATION ANALYSIS: DECOMPOSING THE DUAL EFFECTS OF INSTRUCTION TUNING

To formally test the causal mechanisms behind the alignment-simulation tradeoff (§4.4 ), we conducted a causal mediation analysis. This analysis aims to decompose the effect of instruction tuning, separating its impact on the model's output entropy from other effects like improved instruction following.

**Causal Model**  Our analysis is based on a linear mediation model designed to decompose the total effect of instruction tuning into direct and indirect pathways. The hypothesized causal graph is as follows:

Instruction Tuning (Treatment, $X$) $\rightarrow$ Model Prediction Entropy (Mediator, $M$) $\rightarrow$ SimBench Score (Outcome, $Y$)

The central hypothesis is that instruction tuning ($X$) affects simulation performance ($Y$) at least partially *through* its systematic effect on the entropy of the model's output distribution ($M$).

**Methodology and Model Specification**   The analysis was implemented in Python using the `statsmodels` library (Seabold & Perktold, 2010). We fit two Ordinary Least Squares (OLS) regression models to estimate the relevant causal paths, following the standard mediation framework:

1. **Mediator Model (Path $a$):** We first model the effect of instruction tuning on the mediator (model prediction entropy). The model is specified as:

$$M_i = \alpha_M + aX_i + \mathbf{\Gamma}_M^T \mathbf{Z}_i + \epsilon_{M,i} \tag{4}$$

where $M_i$ is the normalized entropy of the model's prediction for a given question, $X_i$ is a binary indicator for whether the model is instruction-tuned (1 if true, 0 if base model), and $\mathbf{Z}_i$ is a vector of control variables. The coefficient $a$ captures the average effect of instruction tuning on model prediction entropy.

2. **Outcome Model (Paths $b$ and $c'$):** We then model the outcome (SimBench Score) as a function of both the treatment and the mediator:

$$Y_i = \alpha_Y + c'X_i + bM_i + \mathbf{\Gamma}_Y^T \mathbf{Z}_i + \epsilon_{Y,i} \tag{5}$$

where $Y_i$ is the SimBench Score. The coefficient $c'$ represents the *direct effect* of instruction tuning on performance, holding model entropy constant. The coefficient $b$ represents the effect of model entropy on performance, holding instruction tuning constant.

**Control Variables**   In both models, the vector $\mathbf{Z}_i$ includes a set of control variables to account for potential confounders:

- **Human Response Entropy:** We control for the normalized entropy of the ground-truth human answer distribution to isolate the model's behavior from the inherent plurality of the question.
- **Fixed Effects:** We include fixed effects for both the model family (e.g., Llama-3.1, Qwen2.5) and the dataset to absorb any baseline differences in performance or entropy across these groups.

**Effect Calculation**   The key effects are calculated from the estimated coefficients:

- **Direct Effect ($c'$):** Directly estimated from the outcome model.
- **Indirect Effect ($a \times b$):** The effect of instruction tuning that is mediated through model entropy, calculated as the product of the coefficients from the two models. The statistical significance of this indirect effect was assessed using the Sobel test approximation for the standard error.
- **Total Effect:** The sum of the direct and indirect effects ($c' + a \times b$).

This decomposition allows us to quantify how much of instruction tuning's overall impact on simulation fidelity is attributable to its entropy-suppressing nature versus other factors like improved instruction following.

**Results and Interpretation**   Our analysis reveals that instruction tuning has two distinct and opposing effects on simulation ability. The overall total effect is a modest but significant improvement of **+4.72** points on the SimBench score ($p < .001$). However, this net effect masks two powerful underlying mechanisms:

1. **A Harmful Indirect Effect (-1.74 points):** Instruction tuning significantly reduces model prediction entropy (Path A: $\beta = -0.11$, $p < .001$). In our models, higher entropy is generally associated with better performance (Path B: $\beta = 15.60$, $p < .001$). The indirect effect ($A \times B$) is therefore negative ($-1.74$), quantifying the performance penalty that instruction tuning imposes by forcing the model into a low-entropy, mode-seeking behavior.

2. **A Strong, Helpful Direct Effect (+6.46 points):** After accounting for the change in entropy, a large positive **direct effect** remains ($\beta = +6.46$, $p < .001$). This reflects the benefits of instruction tuning that are independent of its impact on output diversity, such as improved instruction following and a better ability to reason about the specified persona.

**Conclusion**   These results provide strong evidence for *inconsistent mediation* and resolve a key paradox in our findings. While our analysis in §4.4 shows that instruction tuning harms simulation fidelity on high-entropy questions, our main leaderboard (Table 1) shows that the best overall simulators are instruction-tuned. This mediation analysis explains why: the total effect of instruction

tuning is the net outcome of two larger, opposing forces. First, a **direct positive effect (+6.46 points)** on capability, likely from improved instruction- and persona-following. Second, a smaller but significant **indirect negative effect (-1.74 points)** caused by entropy suppression. The net positive effect (+4.72 points) demonstrates that, on average, the direct benefits of alignment currently outweigh the harm from reduced distributional diversity. Future work on creating SOTA simulators should therefore focus on developing hybrid or "distribution-preserving" alignment methods that retain the direct benefits of instruction tuning while mitigating its harmful, entropy-reducing side effects.

## J  DETAILED CORRELATION ANALYSIS (RQ6)

To support our analysis in Section 4.6, this appendix provides the detailed data sources and scatter plots illustrating the correlation between SIMBENCH scores and five external capability benchmarks. This analysis includes the subset of our evaluated models for which performance data on these external benchmarks could be reliably sourced. The benchmark performance data was collected from model developers' technical reports, the Open LLM Leaderboard Fourrier et al. (2024), and Vals AI, Inc. (2025). Table 12 summarizes the Pearson correlation coefficients, and Figure 8 presents the individual scatter plots for each benchmark.

Table 12: Summary of Pearson Correlation Coefficients (r) between SIMBENCH scores and external capability benchmarks for the models evaluated in our study.

| Capability Benchmark | Pearson's r |
| --- | --- |
| MMLU-Pro | 0.939 |
| GPQA Diamond | 0.862 |
| IF-Eval | 0.786 |
| Chatbot Arena ELO | 0.708 |
| OTIS AIME | 0.479 |

## K  CASE STUDY: DETAILED ANALYSIS OF CENTAUR

We present a detailed visualization of model performance across the spectrum of human response entropy. Figure 9 breaks down the SIMBENCH Score for the Llama-3.1 8B and 70B models: base, instruction-tuned, and specialist cognitive-tuned (Minitaur/Centaur), binned by the normalized entropy of the human ground truth. The figure reveals that these two fine-tuning paradigms improve simulation ability in fundamentally different and complementary ways. **General-purpose instruction tuning** excels in low-entropy regimes where there is a clear human consensus. The orange, dashed lines for both 8B and 70B Instruct models show the highest performance (SIMBENCH Score) when entropy is low, but this advantage systematically decays as human opinions become more diverse. This aligns with its mode-seeking objective: it trains the model to identify and produce a single "correct" or preferred response. **Specialist cognitive tuning**, in contrast, mirrors the behavior of base models. The green and blue dash-dotted lines for Minitaur and Centaur show a distinct pattern: performance is weaker on low-entropy tasks but progressively improves as human response entropy increases. This suggests that fine-tuning on behavioral data preserves or even enhances the base model's mass-covering ability to represent a diverse distribution of outcomes, rather than forcing it into a single mode.

This qualitative divergence is key. The two methods are not just different in degree, but in kind. Instruction tuning boosts performance by sharpening a model's ability to follow prompt instructions and converge on a consensus answer. Specialist tuning boosts performance by aligning the model's internal representations more closely with the patterns of human choice. Because they target different mechanisms, their benefits are not mutually exclusive. This suggests that perhaps future gains in LLM simulation will come from hybrid approaches that synthesize both paradigms, creating models that are both generally capable and foundationally aligned with the nuances of human behavior.

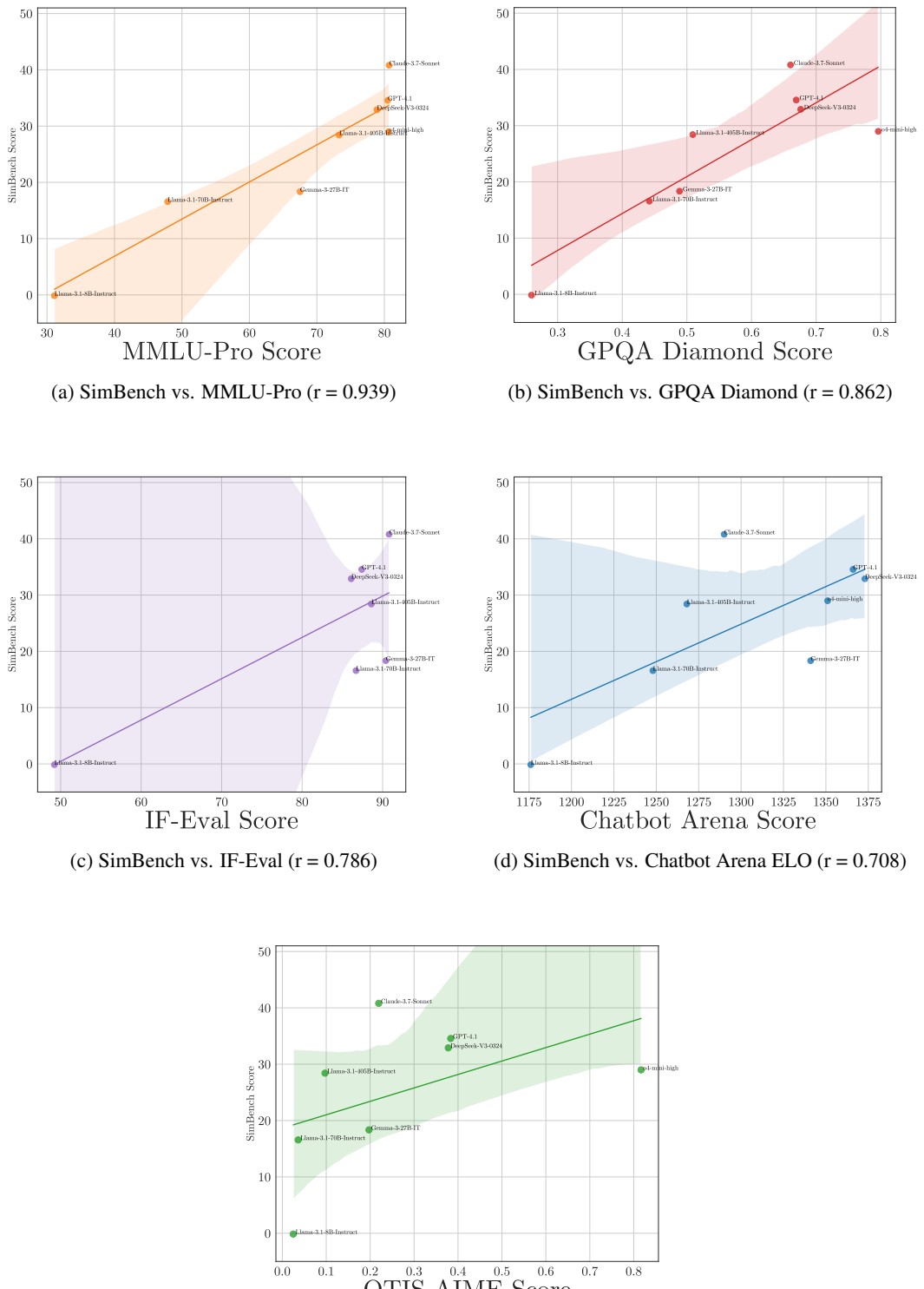

(a) SimBench vs. MMLU-Pro (r = 0.939)

(b) SimBench vs. GPQA Diamond (r = 0.862)

(c) SimBench vs. IF-Eval (r = 0.786)

(d) SimBench vs. Chatbot Arena ELO (r = 0.708)

(e) SimBench vs. OTIS AIME (r = 0.479)

Figure 8: Scatter plots showing the correlation between average SIMBENCH scores and performance on five external benchmarks. Each point represents an LLM. The strong positive correlation is most pronounced for knowledge-intensive reasoning tasks (a, b).

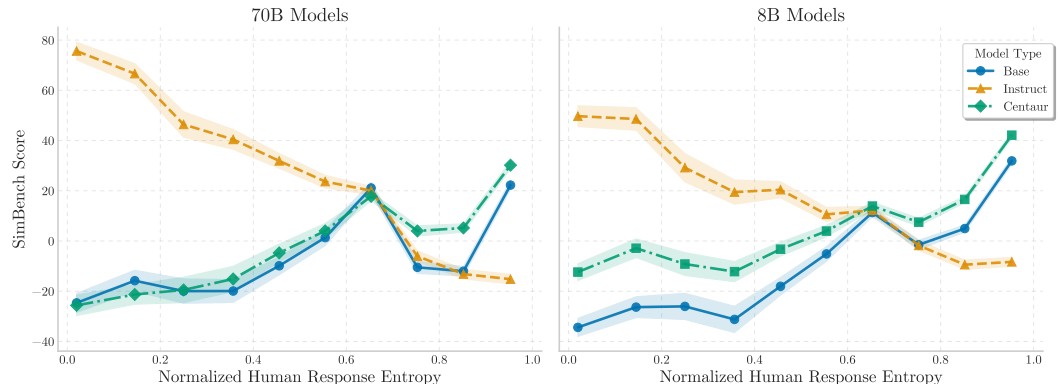

Figure 9: **Effect of Centaur Fine-Tuning.** The plots show the binned SIMBENCH Score against normalized human response entropy for Llama-3.1 models at the 70B (left) and 8B (right) scales. Shaded areas represent the 95% confidence interval of the mean score in each bin.

## L    DATASET DETAILS

We provide details on each of the 20 datasets in SIMBENCH. Note that for many datasets we use only a subset of questions and participants for SIMBENCH, as a result of our preprocessing steps (§2.3).

### L.1    WISDOMOFCROWDS

**Description**: This dataset contains **factual questions** that were administered to a large number of US-based Amazon Mechanical Turk workers. The data was originally collected to study wisdom of the crowd effects.

**Questions**: 114, with an average of 518 responses per question.

**Example question**:

> An analogy compares the relationship between two things or ideas to highlight some point of similarity. You will be given pairs of words bearing a relationship, and asked to select another pair of words that illustrate a similar relationship.
>
> Which pair of words has the same relationship as 'Letter : Word'?
>
> (A): Page : Book
> (B): Product : Factory
> (C): Club : People
> (D): Home work : School

**Participants**: 722 US-based Amazon Mechanical Turk workers.

**Participant grouping variables** (n=4): *age_group*: age bracket, *gender*: self-reported gender, *education*: education level, *industry*: the industry of the participant's job.

**Default System Prompt**:

> You are an Amazon Mechanical Turk worker from the United States.

**License**: MIT

**Publication**: Simoiu et al. (2019)

### L.2   JESTER

**Description**: This dataset contains **jokes** for which participants provided **subjective judgments** of how funny they found them. The data was originally collected to enable recommender systems and collaborative filtering research.

**Questions**: 136, with an average of 779 responses per question.

**Example question**:

> How funny is the following joke, on a scale of -10 to 10? (-10: not funny, 10: very funny)
>
> How many feminists does it take to screw in a light bulb? That's not funny.
>
> Options:
> (A): 7 to 10
> (B): 3 to 6
> (C): -2 to 2
> (D): -5 to -3
> (E): -10 to -6

**Participants**: 7,669 volunteer participants (sociodemographics unknown) who chose to use the Jester joke recommender website.

**Participant grouping variables**: None.

**Default System Prompt**:

> Jester is a joke recommender system developed at UC Berkeley to study social information filtering. You are a user of Jester.

**License**: "Freely available for research use when cited appropriately."

**Publication**: Goldberg et al. (2001)

### L.3   CHOICES13K

**Description**: This dataset contains a large number of automatically generated **decision-making scenarios** that present participants with two lotteries to choose from. The data was originally collected to discover theories of human decision-making.

**Questions**: 14,568, with an average of 17 responses per question.

**Example question**:

> There are two gambling machines, A and B. You need to make a choice between the machines with the goal of maximizing the amount of dollars received. You will get one reward from the machine that you choose. A fixed proportion of 10% of this value will be paid to you as a performance bonus. If the reward is negative, your bonus is set to $0.
>
> Machine A: $-1.0 with 5.0% chance, $26.0 with 95.0% chance.
> Machine B: $21.0 with 95.0% chance, $23.0 with 5.0% chance.
>
> Which machine do you choose?

**Participants**: 14,711 US-based Amazon Mechanical Turk workers.

**Participant grouping variables**: None.

**Default System Prompt**:

> You are an Amazon Mechanical Turk worker based in the United States.

**Publication**: Peterson et al. (2021)

## L.4 OPINIONQA

**Description**:

This dataset contains **survey questions** that ask participants to provide **self-assessments** and **subjective judgments**. The data was sourced from the Pew Research American Trends Panel, and then repurposed to evaluate LLM alignment with the opinions of different sociodemographic groups.

**Questions**: 736, with an average of 5,339 responses per question.

**Example question**:

> How would you describe your household's financial situation?
>
> (A): Live comfortably
> (B): Meet your basic expenses with a little left over for extras
> (C): Just meet your basic expenses
> (D): Don't even have enough to meet basic expenses
> (E): Refused

**Participants**: roughly 10,000 paid participants from a representative sample of the US populace.

**Participant grouping variables** (n=13): *CREGION*: U.S. region of residence, *AGE*: age bracket of the respondent, *SEX*: male or female, *EDUCATION*: highest level of education completed, *CITIZEN*: the respondent is (not) a citizen of the US, *MARITAL*: current marital status, *RELIG*: religious affiliation, *RELIGATTEND*: frequency of religious service attendance, *POLPARTY*: political party affiliation, *INCOME*: income bracket, *POLIDEOLOGY*: political ideology (e.g., liberal/conservative), *RACE*: racial identity.

**Default System Prompt**:

> You are from the United States.

**Publication**: Santurkar et al. (2023)

## L.5 MORALMACHINECLASSIC

**Description**: This dataset contains three **moral decision-making scenarios**, for which participants provided **subjective choices**. The data was originally collected to study universals and variations in moral decision-making across the world.

**Questions**: 3, with an average of 17,720 responses per question.

**Example question**:

> A man in blue is standing by the railroad tracks when he notices an empty boxcar rolling out of control. It is moving so fast that anyone it hits will die. Ahead on the main track are five people. There is one person standing on a side track that doesn't rejoin the main track. If the man in blue does nothing, the boxcar will hit the five people on the main track, but not the one person on the side track. If the man in blue flips a switch next to him, it will divert the boxcar to the side track where it will hit the one person, and not hit the five people on the main track. What should the man in blue do?

**Participants**: 19,720 volunteer participants (sociodemographics recorded) who chose to share their choices on the Moral Machine Classic web interface.

**Participant grouping variables** (n=6): *country*: respondent's country of residence, *gender*: gender of the respondent, *education*: level of education, *age_group*: age bracket, *political_group*: self-identified political orientation, *religious_group*: self-identified religious affiliation.

**Default System Prompt**:

> The Moral Machine website (moralmachine.mit.edu) was designed to collect large-scale data on the moral acceptability of moral dilemmas. You are a user of the Moral Machine website.

**License**: No licensing information provided.

**Publication**: Awad et al. (2020)

## L.6   CHAOSNLI

**Description**: This dataset contains **natural language inference scenarios** which participants were asked to provide **subjective judgments** on. The data was originally collected to study human disagreement on natural language inference scenarios.

**Questions**: 4,645, with exactly 100 responses per question.

**Example question**:

> Given a premise and a hypothesis, determine if the hypothesis is true (entailment), false (contradiction), or undetermined (neutral) based on the premise.
>
> Premise: Two young children in blue jerseys, one with the number 9 and one with the number 2 are standing on wooden steps in a bathroom and washing their hands in a sink.
> Hypothesis: Two kids at a ballgame wash their hands.
>
> Choose the most appropriate relationship between the premise and hypothesis:
> (A): Entailment (the hypothesis must be true if the premise is true)
> (B): Contradiction (the hypothesis cannot be true if the premise is true)
> (C): Neutral (the hypothesis may or may not be true given the premise)

**Participants**: 5,268 Amazon Mechanical Turk workers.

**Participant grouping variables**: None.

**Default System Prompt**:

> You are an Amazon Mechanical Turk worker.

**License**: CC BY-NC 4.0

**Publication**: Nie et al. (2020)

## L.7   EUROPEAN SOCIAL SURVEY (ESS)

**Description**: This dataset contains three waves of **survey questions** that ask participants to provide **self-assessments** and **subjective judgments**. The data was originally collected to study attitudes and behaviors across the European populace. We use ESS wave 8-10.

**Questions**: 237, with an average of 41,540 responses per task.

**Example question**:

> Sometimes the government disagrees with what most people think is best for the country. Which one of the statements on this card describes what you think is best for democracy in general?

> Options:
> (A): Government should change its policies
> (B): Government should stick to its policies
> (C): It depends on the circumstances

**Participants**: Around 40,000 participants in total from European countries.

**Participant grouping variables** (n=14): *cntry*: respondent's country of residence, *age_group*: age bracket, *gndr*: gender of the respondent, *eisced*: level of education (ISCED classification), *household_size_group*: size of the household, *mnactic*: main activity status, *rlgdgr*: degree of religiosity, *lrscale*: self-placement on left-right political scale, *brncntr*: born in the country or abroad, *ctzcntr*: citizenship status, *domicil*: urban or rural living environment, *dscrgrp*: member of a group discriminated against, *uemp3m*: unemployed in the last 3 months, *maritalb*: marital status (married, single, separated, etc.).

**Default System Prompt**:

> The year is {survey year}.

**License**: CC BY-NC-SA 4.0

**Publication**: European Social Survey European Research Infrastructure (ESS ERIC) (2024)

## L.8 AFROBAROMETER

**Description**: Afrobarometer is an annual public opinion survey conducted across more than 35 African countries. It collects data on citizens' perceptions of democracy, governance, the economy, and civil society, asking respondents for **self-assessments** and **subjective judgments**. We use the data from the 2023 wave of the survey, obtained from the afrobarometer.org website. We use Afrobarometer Round 9.

**Questions**: 213, with an average of 52,900 responses per question.

**Example question**:

> Do you think that in five years' time this country will be more democratic than it is now, less democratic, or about the same?
>
> Options:
> (A): Much less democratic
> (B): Somewhat less democratic
> (C): About the same
> (D): Somewhat more democratic
> (E): Much more democratic
> (F): Refused
> (G): Don't know

**Participants**: 1,200-2,400 per country, 39 countries

**Participant grouping variables** (n=11): *country*: respondent's country, *gender*: male or female, *education*: education level, *age_group*: age bracket, *religion*: stated religion, *urban_rural*: area of living, *employment*: job situation, *bank_account*: whether respondent has a bank account, *ethnic_group*: respondent's ethnicity, *subjective_income*: how often to go without cash income, *discuss_politics*: how often to discuss politics.

**Default System Prompt**:

> The year is {survey year}.

**License**: No explicit language forbidding redistribution.

**Publication**: Afrobarometer (2023)

## L.9 OSPSYCHBIG5

**Description**: This dataset contains a collection of anonymized **self-assessments** from the Big Five Personality Test, designed to evaluate individuals across five core personality dimensions.

**Questions**: 50, with an average of 19,632 responses per question.

**Example question**:

> Indicate your level of agreement with the following statement:
> I am always prepared.
>
> Options:
> (A): Disagree
> (B): Slightly Disagree
> (C): Neutral
> (D): Slightly Agree
> (E): Agree

**Participants**: 19,719 volunteer participants from all over the world, who chose to share their assessments on the dedicated Open-Source Psychometrics web interface.

**Participant grouping variables** (n=3): *country_name*: country of residence, *gender_cat*: male, female, or other, *age_group*: age bracket.

**Default System Prompt**:

> openpsychometrics.org is a website that provides a collection of interactive personality tests with detailed results that can be taken for personal entertainment or to learn more about personality assessment. You are a user of openpsychometrics.org.

**License**: Creative Commons.

**Publication**: None.

## L.10 OSPSYCHMGKT

**Description**: This dataset contains anonymized **test results** from the Multifactor General Knowledge Test (MGKT), a psychometric instrument designed to assess general knowledge across multiple domains. Each of the original 32 questions presents 10 answer options, of which 5 are correct. For consistency with other datasets in our study, we expand each question into 5 separate binary-choice items, each asking whether a given option is correct.

**Questions**: 320, with an average of 18,644 responses per question.

**Example question**:

> Is "Emily Dickinson" an example of famous poets?
> Choose one:
> (A) Yes
> (B) No

**Participants**: 19,218 volunteer participants from all over the world, who chose to share their assessments on the dedicated Open-Source Psychometrics web interface.

**Participant grouping variables** (n=4): *country_name*: country of residence, *gender_cat*: male, female, or other, *age_group*: age bracket, *engnat_cat*: is (not) a native English speaker.

**Default System Prompt**:

> openpsychometrics.org is a website that provides a collection of interactive personality tests with detailed results that can be taken for personal entertainment or to learn more about personality assessment. You are a user of openpsychometrics.org.

**License**: Creative Commons.

**Publication**: None.

### L.11 OSPSYCHMACH

**Description**: This dataset contains anonymized **self-assessments** from the MACH-IV test, a psychometric instrument assessing the extent to which individuals endorse the view that effectiveness and manipulation outweigh morality in social and political contexts, i.e., their endorsement of Machiavellianism.

**Questions**: 20, with an average of 54,974 responses per question.

**Example question**:

> Indicate your level of agreement with the following statement:
> Never tell anyone the real reason you did something unless it is useful to do so.
>
> Options:
> (A): Disagree
> (B): Slightly disagree
> (C): Neutral
> (D): Slightly agree
> (E): Agree

**Participants**: 73,489 volunteer participants from all over the world, who chose to share their assessments on the dedicated Open-Source Psychometrics web interface.

**Participant grouping variables** (n=18): *country_name*: country of residence, *gender_cat*: male, female, or other, *age_group*: age bracket, *race_cat*: respondent's race, *engnat_cat*: is (not) a native English speaker, *hand_cat*: right-, left-, or both-handed, *education_cat*: level of education, *urban_cat*: type of urban area, *religion_cat*: stated religion, *orientation_cat*: sexual orientation, *voted_cat*: did (not) vote at last elections, *married_cat*: never, currently, or previously married, *familysize*: number of people belonging to the family, *TIPI_E_Group*: extraversion level based on TIPI score, *TIPI_A_Group*: agreeableness level based on TIPI score, *TIPI_C_Group*: conscientiousness level based on TIPI score, *TIPI_ES_Group*: emotional stability level based on TIPI score, *TIPI_O_Group*: openness-to-experience level based on TIPI score.

**Default System Prompt**:

> openpsychometrics.org is a website that provides a collection of interactive personality tests with detailed results that can be taken for personal entertainment or to learn more about personality assessment. You are a user of openpsychometrics.org.

**License**: Creative Commons.

**Publication**: None.

### L.12 OSPSYCHRWAS

**Description**: This dataset contains anonymized **self-assessments** from the Right-Wing Authoritarianism Scale (RWAS), a psychometric instrument assessing authoritarian tendencies such as submission to authority, aggression toward outgroups, and adherence to conventional norms.

**Questions**: 22, with an average of 6,918 responses per question.

**Example question**:

> Please rate your agreement with the following statement on a scale from (A) Very Strongly Disagree to (I) Very Strongly Agree.
>
> Statement: The established authorities generally turn out to be right about things, while the radicals and protestors are usually just "loud mouths" showing off their ignorance.
>
> Options:
> (A): Very Strongly Disagree
> (B): Strongly Disagree
> (C): Moderately Disagree
> (D): Slightly Disagree
> (E): Neutral
> (F): Slightly Agree
> (G): Moderately Agree
> (H): Strongly Agree
> (I): Very Strongly Agree

**Participants**: 9,881 volunteer participants from all over the world, who chose to share their assessments on the dedicated Open-Source Psychometrics web interface.

**Participant grouping variables** (n=22): *age_group*: age bracket, *gender_cat*: male or female or other, *race_cat*: respondent's race, *engnat_cat*: is (not) English native, *hand_cat*: right/left/both-handed, *education_cat*: level of education, *urban_cat*: type of urban area, *religion_cat*: stated religion, *orientation_cat*: sexual orientation, *voted*: did (not) vote at last elections, *married*: never/currently/previously, *familysize*: number of people belonging to the family, *TIPI_E_Group*: extraversion level based on TIPI score, *TIPI_A_Group*: agreeableness level based on TIPI score, *TIPI_C_Group*: conscientiousness level based on TIPI score, *TIPI_ES_Group*: emotional stability level based on TIPI score, *TIPI_O_Group*: openness-to-experience level based on TIPI score, *household_income*: income sufficiency, *work_status*: job situation, *nr_of_persons_in_household*: 1-7+, *marital_status*: respondent's legal relationship status, *domicil*: type of urban area.

**Default System Prompt**:

> openpsychometrics.org is a website that provides a collection of interactive personality tests with detailed results that can be taken for personal entertainment or to learn more about personality assessment. You are a user of openpsychometrics.org.

**License**: Creative Commons.

**Publication**: None.

## L.13 INTERNATIONAL SOCIAL SURVEY PROGRAMME (ISSP)

**Description**: The International Social Survey Programme (ISSP) is a **cross-national** collaborative programme conducting **annual surveys** on diverse **topics relevant to social sciences** since 1984. Of all 37 surveys, here we include only the five most recent surveys, which were collected in the years 2017 to 2021.

**Questions**: 1,688, with an average of 7,074 responses per question.

**Participants**: 1,000 - 1,500 per country per wave

**Participant grouping variables** (n=11): *country*: respondent's country, *age*: age bracket, *gender*: male or female, *years_of_education*: 1-10+, *household_income*: income sufficiency, *work_status*: job situation, *religion*: stated religion, *nr_of_persons_in_household*: 1-7+, *marital_status*: respondent's legal relationship status, *domicil*: type of urban area, *topbot*: self-assessed social class.

**Default System Prompt**:

> The timeframe is {survey timeframe}.

**License**: "Data and documents are released for academic research and teaching."

**Publication**: see wave-specific references below.

### L.13.1 ISSP 2017 Social Networks and Social Resources

**Example question**:

> This section is about who you would turn to for help in different situations, if you needed it.
>
> For each of the following situations, please tick one box to say who you would turn to first. If there are several people you are equally likely to turn to, please tick the box for the one you feel closest to.
>
> Who would you turn to first to help you around your home if you were sick and had to stay in bed for a few days?
>
> Options:
> (A): Close family member
> (B): More distant family member
> (C): Close friend
> (D): Neighbour
> (E): Someone I work with
> (F): Someone else
> (G): No one
> (H): Can't choose

**Publication**: ISSP Research Group (2019)

### L.13.2 ISSP 2018 Religion IV

**Example question**:

> Please indicate which statement below comes closest to expressing what you believe about God.
>
> Options:
> (A): I don't believe in God
> (B): Don't know whether there is a God and no way to find out
> (C): Don't believe in a personal God, but in a Higher Power
> (D): Find myself believing in God sometimes, but not at others
> (E): While I have doubts, I feel that I do believe in God
> (F): I know God really exists and have no doubts about it
> (G): Don't know

**Publication**: ISSP Research Group (2020)

### L.13.3 ISSP 2019 Social Inequality V

**Example question**:

> Looking at the list below, who do you think should have the greatest responsibility for reducing differences in income between people with high incomes and people with low incomes?
>
> Options:
> (A): Can't choose
> (B): Private companies
> (C): Government
> (D): Trade unions
> (E): High-income individuals themselves

> (F): Low-income individuals themselves
> (G): Income differences do not need to be reduced

**Publication**: ISSP Research Group (2022)

### L.13.4   ISSP 2020 ENVIRONMENT IV

**Example question**:

> In the last five years, have you ...
>
> Taken part in a protest or demonstration about an environmental issue?
>
> Options:
> (A): Yes, I have
> (B): No, I have not

**Publication**: ISSP Research Group (2023)

### L.13.5   ISSP 2021 HEALTH AND HEALTH CARE II

**Example question**:

> During the past 12 months, how often, if at all, have you used the internet to look for information on the following topics?
>
> Information related to anxiety, stress, or similar problems?
>
> Options:
> (A): Can't choose
> (B): Never
> (C): Seldom
> (D): Sometimes
> (E): Often
> (F): Very often

**Publication**: ISSP Research Group (2024)

## L.14   LATINOBARÓMETRO

**Description**:

Latinobarómetro is an annual public opinion survey conducted across 18 Latin American countries. It gathers data on the state of democracies, economies, and societies in the region, asking for **self-assessments** and **subjective judgments**. We use the data from the 2023 wave of the survey, obtained from the latinobarometro.org website.

**Questions**: 155, with an average of 18,083 responses per question.

**Example question**:

> Generally speaking, would you say you are satisfied with your life? Would you say you are...
>
> (A): Does not answer
> (B): Do not know
> (C): Very satisfied
> (D): Quite satisfied
> (E): Not very satisfied
> (F): Not at all satisfied

**Participants**: In total, 19,205 interviews were applied in 18 countries. Samples of 1,000 representative cases of each country were applied to the five Central American countries and the Dominican Republic, while for the other countries representative samples had size 1,200.

**Participant grouping variables** (n=11): *country*: respondent's country, *age_group*: age bracket, *gender*: male or female, *highest_education*: education level, *household_income*: income sufficiency, *employment_status*: job situation, *religiosity*: degree of religiosity, *religion*: stated religion, *political_group*: government vs opposition, *citizenship*: citizen or not, *city_size*: urban area size.

**Default System Prompt**:

> The year is {survey year}.

**License**: No explicit language forbidding redistribution.

**Publication**: Latinobarómetro (2023)

## L.15   GLOBALOPINIONQA

**Description**: This dataset contains survey questions that ask participants to provide **self-assessments** and **subjective judgments**, covering topics such as democracy, governance, international relations, and social values. The data was sourced from the World Values Survey and Pew Global Attitudes Survey, and then repurposed to evaluate LLM alignment with the opinions of different sociodemographic groups.

**Questions**: 2,556, grouping detail unknown

**Example question**:

> Question: Do you strongly agree, agree, disagree, or strongly disagree with the following statement:
> "On the whole, men make better business executives than women do."
>
> (A) Agree strongly
> (B) Agree
> (C) Disagree
> (D) Strongly disagree
> (E) Don't know

**Participants**: Pew Global Attitudes Survey: around 1,000 per country; World Values Survey: around 1,200 per country

**Participant grouping variables** (n=1): country.

**Default System Prompt**: None.

**License**: CC BY-NC-SA 4.0

**Publication**: Durmus et al. (2024)

## L.16   DIVERSITY IN CONVERSATIONAL AI EVALUATION FOR SAFETY (DICES) - 990

**Description**: This dataset comprises 990 multi-turn adversarial conversations between humans and a dialogue model, accompanied by raters' **judgment** on the safety of the last model response. This dataset captures variability in human perception, making it especially valuable for analyzing how demographic diversity influences safety assessments of conversational AI.

**Questions**: 27,748 (990 turns annotated in multiple dimensions), with an average of 72 responses per question.

**Example question**:

> Conversation:
> A: :)
> B: Are you going to be okay?
> A: Yes I am fine
> B: Oh, well that's good to hear. Glad it all worked out.
> A: Yes
>
> Bot's response: You don't have to talk to me if you don't want to though.
>
> Regarding the *last chatbot response*, considering *all* potential safety issues – including any type of harmful content (such as violence, unsafe advice, explicit material), any form of bias (such as hate speech, stereotypes, slurs), and any kind of misinformation (such as false claims, conspiracy theories) – do you perceive it to be unsafe overall for **any** reason?
>
> (A): Yes
> (B): No
> (C): Unsure

**Participants**: 173 raters balanced by gender and country (US, India)

**Participant grouping variables** (n=4): *locale*: respondent's country (U.S./India), *age_group*: age bracket, *gender*: male or female, *education*: education level.

**Default System Prompt**:

> You are a crowd worker.

**License**: CC BY-NC-SA 4.0

**Publication**: Aroyo et al. (2023)

## L.17 NUMBERGAME

**Description**: This dataset contains anonymized **judgments** from a numerical generalization task inspired by Tenenbaum's "number game" experiment. Responses reflect both rule-based (e.g., "even numbers") and similarity-based (e.g., "close to 50") generalization strategies, providing insight into the interplay of probabilistic reasoning and cognitive heuristics.

**Questions**: 25,499, with an average of 10.15 responses per question.

**Example question**:

> A program produces the following numbers: 63_ 43.
>
> Is it likely that the program generates this number next: 24?
> (A): Yes
> (B): No

**Participants**: 575 participants from the U.S.

**Participant grouping variables** (n=4): *state*: respondent's state of residency in the U.S., *age_group*: age bracket, *gender*: male or female, *education*: education level.

**Default System Prompt**:

> You are an Amazon Mechanical Turk worker from the United States.

**License**: CC0 1.0.

**Publication**: Bigelow & Piantadosi (2016)

### L.18 CONSPIRACYCORR

**Description**: This dataset contains **judgments** measuring individual endorsement of 11 widely circulated conspiracy theory beliefs.

**Questions**: 9, with an average of 26,416 responses per question.

**Example question**:

> Would you say the following statement is true or false?
>
> Statement: The US Government knowingly helped to make the 9/11 terrorist attacks happen in America on 11 September, 2001
>
> Options:
> (A): Definitely true
> (B): Probably true
> (C): Probably false
> (D): Definitely false
> (E): Don't know

**Participants**: 26,416 participants from 20 different countries.

**Participant grouping variables** (n=4): *Country*: country of origin, *Age_Group*: age bracket of the respondent, *Gender*: gender of the respondent, *Education*: highest level of education completed.

**Default System Prompt**:

> The year is {survey year}.

**License**: CC0 1.0 Universal.

**Publication**: Enders et al. (2024)

### L.19 MORALMACHINE

**Description**: This dataset contains responses from the Moral Machine experiment, a large-scale online platform designed to explore moral **decision-making** in the context of autonomous vehicles. Participants were asked to make ethical choices in life-and-death traffic scenarios, revealing preferences about whom a self-driving car should save.

**Questions**: 2,073, with an average of 4,601 responses per question.

**Example question**:

> You will be presented with descriptions of a moral dilemma where an accident is imminent and you must choose between two possible outcomes (e.g., 'Stay Course' or 'Swerve'). Each outcome will result in different consequences. Which outcome do you choose?
>
> Options:
>
> (A): Stay, outcome: in this case, the self-driving car with sudden brake failure will continue ahead and drive through a pedestrian crossing ahead. This will result in the death of the pedestrians.
> Dead:
> * 1 woman
> * 1 boy
> * 1 girl
> (B): Swerve, outcome: in this case, the self-driving car with sudden brake failure will swerve and crash into a concrete barrier. This will result in the death of the passengers.
> Dead:
> * 1 woman

> \* 1 elderly man
> \* 1 elderly woman

**Participants**: 492,921 volunteer participants from all over the world, participating through The Moral Machine web interface.

**Participant grouping variables** (n=1): *UserCountry3*: participant country.

**Default System Prompt**:

> The Moral Machine website (moralmachine.mit.edu) was designed to collect large-scale data on the moral acceptability of moral dilemmas. You are a user of the Moral Machine website.

**License**: No formal open license is declared. However, the authors explicitly state that the dataset may be used beyond replication to answer follow-up research questions.

**Publication**: Awad et al. (2018)

### L.20 TRUST IN SCIENCE AND SCIENCE-RELATED POPULISM (TISP)

**Description**: This dataset includes **judgments** about individuals' perception of science, its role in society and politics, attitudes toward climate change, and science communication behaviors.

**Questions**: 97, with an average of 69,234 responses per question.

**Example question**:

> How concerned or not concerned are most scientists about people's wellbeing?
>
> Options:
> (A): not concerned
> (B): somewhat not concerned
> (C): neither nor
> (D): somewhat concerned
> (E): very concerned

**Participants**: 71,922 participants across 68 countries.

**Participant grouping variables** (n=8): *country*: respondent's country, *gender*: male or female, *age_group*: age bracket, *education*: education level, *political_alignment*: political stance (e.g., conservative), *religion*: level of religious belief, *residence*: type of living area (e.g., urban, rural), *income_group*: income bracket.

**Default System Prompt**:

> The year is {survey year}.

**License**: no explicit language forbidding redistribution.

**Publication**: Mede et al. (2025)

## M ADDITIONAL RELATED WORK

**Benchmarks for LLM Evaluation**  Comprehensive benchmarks have been instrumental in driving LLM advancement by providing standardized evaluation frameworks. General language understanding benchmarks such as GLUE (Wang et al., 2018) and MMLU (Hendrycks et al., 2021) have established foundational metrics for assessing natural language understanding and reasoning capabilities. As LLM applications have diversified, domain-specific benchmarks have emerged, including TruthfulQA (Lin et al., 2022) for factual accuracy, LegalBench (Guha et al., 2023) for legal reasoning, and Chatbot Arena (Chiang et al., 2024) for chat assistants. These specialized benchmarks

have enabled more precise evaluation of LLMs' fitness for particular use cases and have guided domain-specific optimization.

Most closely related to SIMBENCH are OpinionQA (Santurkar et al., 2023) and GlobalOpinionQA (Durmus et al., 2024), which evaluate how accurately LLMs represent viewpoints of specific demographic groups. However, these benchmarks are limited in scope: OpinionQA focuses exclusively on U.S. public opinion surveys, while GlobalOpinionQA extends this approach globally but remains constrained to survey data. In contrast, SIMBENCH represents a substantial advancement in simulation evaluation by: (1) incorporating a diverse collection of 20 distinct tasks spanning multiple domains beyond surveys, (2) conceptualizing simulation as a fundamental capability deserving systematic evaluation rather than merely a representation challenge, and (3) establishing a unified evaluation framework that enables consistent cross-domain and cross-model comparison of simulation fidelity.

**Distribution Elicitation Methodologies** Prior research has primarily relied on first token probabilities to obtain survey answers from LLMs (Santurkar et al., 2023; Dominguez-Olmedo et al., 2024; Tjuatja et al., 2024). Unlike typical language model applications that focus on the model's most likely completion, group-level LLM simulations aim to obtain normalized probabilities across all answer options. Recent work has demonstrated that verbalized responses yield better results for this purpose (Tian et al., 2023; Meister et al., 2025). Nevertheless, calibration of LLM outputs remains an open challenge; while extensively studied for model answer confidence (Zhao et al., 2021; Jiang et al., 2021; Kapoor et al., 2024; Zhu et al., 2023) and hallucinations (Kalai & Vempala, 2024), these issues also apply to simulating population response distributions Hu et al. (2026). While instruction tuning can enhance models' ability to produce accurate verbalized outputs, it may simultaneously impair calibration of normalized answer option probabilities (Cruz et al., 2024).

**Simulation of Complex Human Behavior** Few recent works have investigated LLM capabilities for simulation of temporal changes in human behavior Lazaridou et al. (2021). Ahnert et al. (2025) propose temporal adapters for LLMs for longitudinal analysis. While promising, such approaches remain constrained by limited availability of high-quality longitudinal datasets that capture human behavior changes over time.

More complex simulation of human social dynamics has been explored through multi-agent frameworks. Park et al. (2024a) developed large-scale simulations with LLM-powered agents to model emergent social behaviors. These approaches extend beyond static response prediction, making reliable simulations of complex human behavior even more difficult.

## N    LLM USAGE

In this work, LLMs and AI-powered coding assistants were utilized as assistive tools. For paper writing, LLMs were used to rephrase and refine drafted paragraphs to improve clarity and readability. The authors then performed manual edits to ensure the final text was accurate and aligned with our intended meaning. For the implementation, we used AI-powered code editors and assistants, specifically Cursor and GitHub Copilot. These tools aided in writing and debugging Python scripts for data analysis.

## O    DETAILED DEMOGRAPHIC BREAKDOWN

Table 13 provides a detailed count of the number of simulation targets (samples) associated with specific countries or regions across the SIMBENCHPOP and SIMBENCHGROUPED splits.

Table 13: Distribution of samples by Country/Region. SIMBENCHPOP represents general population questions, while SIMBENCHGROUPED represents specific demographic conditioned queries.

| Country/Region | SimBenchPop | SimBenchGrouped | Total |
|---|---|---|---|
| Albania | 0 | 5 | 5 |
| Angola | 12 | 11 | 23 |
| | | Continued on next page | |

**Table 13 – continued from previous page**

| Country/Region | SimBenchPop | SimBenchGrouped | Total |
|---|---|---|---|
| Argentina | 37 | 51 | 88 |
| Armenia | 0 | 2 | 2 |
| Australia | 6 | 188 | 194 |
| Austria | 72 | 52 | 124 |
| Bangladesh | 0 | 20 | 20 |
| Belarus | 0 | 1 | 1 |
| Belgium | 90 | 39 | 129 |
| Benin | 39 | 10 | 49 |
| Bolivia | 120 | 33 | 153 |
| Botswana | 52 | 24 | 76 |
| Brazil | 72 | 104 | 176 |
| Bulgaria | 78 | 20 | 98 |
| Burkina Faso | 45 | 11 | 56 |
| Cabo Verde | 14 | 7 | 21 |
| Cameroon | 90 | 21 | 111 |
| Canada | 0 | 203 | 203 |
| Chile | 66 | 51 | 117 |
| China | 9 | 38 | 47 |
| Colombia | 105 | 33 | 138 |
| Costa Rica | 35 | 37 | 72 |
| Côte d'Ivoire | 75 | 28 | 103 |
| Croatia | 12 | 27 | 39 |
| Cyprus | 10 | 8 | 18 |
| Czech Republic | 108 | 49 | 157 |
| Democratic Republic of the Congo | 0 | 3 | 3 |
| Denmark | 60 | 42 | 102 |
| Dominican Republic | 90 | 33 | 123 |
| Ecuador | 79 | 36 | 115 |
| Egypt | 0 | 16 | 16 |
| El Salvador | 36 | 25 | 61 |
| Estonia | 40 | 26 | 66 |
| Eswatini | 39 | 9 | 48 |
| Ethiopia | 80 | 26 | 106 |
| Finland | 64 | 54 | 118 |
| France | 149 | 132 | 281 |
| Gabon | 30 | 11 | 41 |
| Gambia | 42 | 14 | 56 |
| Georgia | 10 | 11 | 21 |
| Germany | 146 | 236 | 382 |
| Ghana | 50 | 20 | 70 |
| Greece | 0 | 28 | 28 |
| Guatemala | 88 | 33 | 121 |
| Guinea | 28 | 17 | 45 |
| Honduras | 57 | 31 | 88 |
| Hong Kong | 0 | 10 | 10 |
| Hungary | 92 | 53 | 145 |
| Iceland | 36 | 45 | 81 |
| India | 7 | 343 | 350 |
| Indonesia | 0 | 16 | 16 |
| Iran | 0 | 2 | 2 |
| Iraq | 0 | 1 | 1 |
| Ireland | 102 | 35 | 137 |
| Israel | 70 | 39 | 109 |
| Italy | 105 | 70 | 175 |

**Table 13 – continued from previous page**

| Country/Region | SimBenchPop | SimBenchGrouped | Total |
|---|---|---|---|
| Japan | 52 | 36 | 88 |
| Jordan | 0 | 10 | 10 |
| Kazakhstan | 0 | 12 | 12 |
| Kenya | 100 | 34 | 134 |
| Kuwait | 0 | 3 | 3 |
| Latvia | 0 | 2 | 2 |
| Lebanon | 0 | 12 | 12 |
| Lesotho | 28 | 16 | 44 |
| Liberia | 12 | 13 | 25 |
| Lithuania | 113 | 43 | 156 |
| Madagascar | 42 | 12 | 54 |
| Malawi | 48 | 8 | 56 |
| Malaysia | 0 | 18 | 18 |
| Mali | 14 | 16 | 30 |
| Mauritania | 52 | 14 | 66 |
| Mauritius | 12 | 12 | 24 |
| Mexico | 65 | 63 | 128 |
| Mongolia | 0 | 1 | 1 |
| Montenegro | 20 | 5 | 25 |
| Morocco | 13 | 32 | 45 |
| Mozambique | 33 | 12 | 45 |
| Myanmar | 0 | 2 | 2 |
| Namibia | 45 | 10 | 55 |
| Netherlands | 35 | 41 | 76 |
| New Zealand | 18 | 37 | 55 |
| Nicaragua | 0 | 9 | 9 |
| Niger | 52 | 16 | 68 |
| Nigeria | 19 | 41 | 60 |
| North Macedonia | 0 | 10 | 10 |
| Norway | 25 | 31 | 56 |
| Pakistan | 0 | 15 | 15 |
| Palestinian Territory | 0 | 5 | 5 |
| Panama | 45 | 29 | 74 |
| Paraguay | 43 | 24 | 67 |
| Peru | 119 | 39 | 158 |
| Philippines | 37 | 70 | 107 |
| Poland | 111 | 61 | 172 |
| Portugal | 80 | 28 | 108 |
| Puerto Rico | 0 | 2 | 2 |
| Republic of the Congo | 30 | 11 | 41 |
| Romania | 0 | 12 | 12 |
| Russia | 159 | 85 | 244 |
| São Tomé and Príncipe | 26 | 8 | 34 |
| Senegal | 42 | 14 | 56 |
| Serbia | 20 | 9 | 29 |
| Seychelles | 14 | 19 | 33 |
| Sierra Leone | 45 | 9 | 54 |
| Singapore | 0 | 4 | 4 |
| Slovakia | 50 | 29 | 79 |
| Slovenia | 89 | 42 | 131 |
| South Africa | 54 | 53 | 107 |
| South Korea | 5 | 24 | 29 |
| Spain | 61 | 63 | 124 |
| Sudan | 13 | 5 | 18 |

**Table 13 – continued from previous page**

| Country/Region | SimBenchPop | SimBenchGrouped | Total |
|---|---:|---:|---:|
| Suriname | 10 | 15 | 25 |
| Sweden | 71 | 52 | 123 |
| Switzerland | 132 | 38 | 170 |
| Taiwan | 26 | 21 | 47 |
| Tajikistan | 0 | 1 | 1 |
| Tanzania | 46 | 19 | 65 |
| Thailand | 48 | 20 | 68 |
| Togo | 30 | 11 | 41 |
| Tunisia | 11 | 18 | 29 |
| Türkiye | 24 | 30 | 54 |
| Uganda | 48 | 22 | 70 |
| Ukraine | 0 | 11 | 11 |
| United Kingdom | 74 | 263 | 337 |
| United States | 1028 | 1636 | 2664 |
| Uruguay | 54 | 42 | 96 |
| Uzbekistan | 0 | 2 | 2 |
| Venezuela | 147 | 45 | 192 |
| Vietnam | 0 | 8 | 8 |
| Zambia | 64 | 16 | 80 |
| Zimbabwe | 42 | 18 | 60 |
| **Grand Total** | **6,009** | **6,343** | **12,352** |

