# OpenReview forum: "SimBench: Benchmarking the Ability of Large Language Models to Simulate Human Behaviors"
_ICLR.cc/2026/Conference — ICLR 2026 Poster_

### Official Review · Reviewer_RbiV · 2025-10-28

**Soundness:** 3
**Presentation:** 4
**Contribution:** 4
**Rating:** 10
**Confidence:** 4

**Summary:**

The paper proposes SimBench, a benchmark for testing whether models can reproduce group-level human response distributions across 20 diverse datasets. They find a log-linear scaling of simulation performance with model size; no consistent benefit from reasoning models; and a trade-off with instruction-tuning where it improves performance on questions that have more consensus and decreases performance on questions that have more diversity. They also replicate past work, such as showing that performance worsens when targeting specific demographic groups.

**Strengths:**

1. There is a great need for high-quality evaluations in the nascent field of LLM social simulations, which this paper targets.
2. The data curation is rigorous, including scale, diversity (of task and participants), and cleaning/consistency.
3. The empirical findings are interesting and well-framed as tentative and qualified—ripe for future research.
4. There is extensive documentation and justification in the appendices, such as tests of alternative metrics.

**Weaknesses:**

1. The baseline is "equal probability to all response options," but does that make sense for real applications? Why not modal responses or a linear predictor? I'm particularly wondering about how to clarify the 40.80/100 score because it seems a more grounded alternative (e.g., the 85% test-retest accuracy in Park et al. 2025) would be useful if possible. I think accuracy (and even correlation) is more informative than TV in this regard.
2. The authors should consider including comparisons to quantitative machine learning models, perhaps with LLM embeddings for text inputs, as a comparison to the LLM social simulations. Knowing how LLMs compare to conventional models is important because that is often the counterfactual for researchers.
3. There is much heterogeneity in the datasets, as the authors acknowledge, but the paper has limited exploration of the different results in these areas and what they imply for simulation. Social science is extremely broad, and it is important to differentiate topics, rigor, elicitation methods (of the humans), etc.
4. A number of weaknesses are acknowledged by the authors but remain nonetheless, such as the reliance on multiple-choice format questions—understandable, but limits applicability to other research formats.

**Questions:**

1. I read the "Training Data Overlap" section, but did the authors empirically address in any way the possibility of models, especially larger and newer models, having the datasets (or the papers resulting from those datasets) in the training data? It would be useful to include tests on unpublished data, as did Hewitt and Ashokkumar et al. (2024; already cited in the paper), or at least foreground this limitation in the main text.
2. The analyses in the paper focus on the SimBench dataset as a whole, but with its diversity, what differences are there in performance across it? I ask in part because the target for simulations is often new studies where the results are highly uncertain. Ideally, one could examine performance on the parts of SimBench that are the best proxies for this (e.g., studies where humans were surprised by the results, studies on less-well-studied topics).

---

> ### Author Response · Authors · 2025-11-19
>
> **Thank you very much for your thoughtful review and for recognizing the need for standardized evaluation in LLM social simulation, the rigor of our data curation, and our interesting, well-framed findings. We appreciate your encouraging assessment and address your specific concerns below.**
>
> ---
> >1.*The baseline is "equal probability to all response options," but does that make sense for real applications? Why not modal responses or a linear predictor? I'm particularly wondering about how to clarify the 40.80/100 score because it seems a more grounded alternative (e.g., the 85% test-retest accuracy in Park et al. 2025) would be useful if possible. I think accuracy (and even correlation) is more informative than TV in this regard.*
>
> **Response:**
>
> We agree that the uniform baseline is simple, but it serves an important methodological purpose: it establishes a true “zero-knowledge” floor for zero-shot simulation. Unlike linear predictors or modal baselines, which require a supervised setting with available training data, the uniform baseline applies to novel scenarios where the answer is unknown a priori.
>
> Regarding the alternative metrics suggested:
>
> *   **Accuracy:** We prioritize TVD because accuracy is **mode-seeking**—it rewards placing 100% probability mass on the majority option, whereas faithful simulation requires capturing the full diversity of opinion (including minority views).
> *   **Correlation:** While useful for trends, correlation ignores absolute probability magnitudes (e.g., `[1%, 2%]` correlates perfectly with `[10%, 20%]`). Simulation requires exact probability matching. We do, however, include Rank Correlation in Appendix F (Table 6) as a robustness check.
>
> Our S score is specifically designed to make performance relative to this floor highly interpretable **for a general readership**: S=0 signifies performance that is no better than this baseline, while S=100 signifies a perfect distributional match. The fact that this seemingly low bar is not met by several evaluated models is a finding in itself: reliable human simulation is a non-trivial capability, and many current (small) models are not just inaccurate, but can be actively misleading compared to a simple uniform guess.
>
> Additionally, for readers **interested in the underlying metric**, we provide the raw TVD scores in Table 10. The top-performing model achieves an average TVD of 0.191. To make this more grounded, as the reviewer suggests, this score means that the model has, on average, misallocated 38.2% (i.e., 2 × 0.191) of its total probability mass compared to the human ground truth. This provides a concrete quantitative measure of the gap that still exists between today’s best models and a truly faithful human simulator.
>
> ---
>
> >2.*The authors should consider including comparisons to quantitative machine learning models, perhaps with LLM embeddings for text inputs, as a comparison to the LLM social simulations. Knowing how LLMs compare to conventional models is important because that is often the counterfactual for researchers.*
>
> **Response:**
>
> We completely agree that understanding how generative LLM simulators compare to conventional supervised models is very important. While this direct comparison is outside the scope of this study, SimBench provides the infrastructure to enable this inquiry in future work.
>
> * Our paper's primary goal is to benchmark the innate, **zero-shot** simulation capability of LLMs. This paradigm, of generating a distributional prediction for any novel question without task-specific training, is a core promise of LLMs as simulators. Including a comparison to a supervised model, such as a classifier over LLM embeddings, would introduce a fundamentally different simulation paradigm.
>
> * This would shift the focus from a model's intrinsic simulation abilities to its capacity to learn a specific prediction task from a training set. While a valid and important question, it is a distinct one from what we set out to measure. Additionally, the immense flexibility of the generative approach is precisely why LLM-based simulation has become so popular. An LLM can be prompted to produce outputs in virtually any format—multiple-choice, Likert scales, rankings, or even free-text justifications.
>
> * Furthermore, this comparison would require significant methodological choices that are themselves open research questions. For example: How would one fairly create training/test splits across 20 diverse datasets without creating information leakage? How would a single supervised model be architected to handle the variable number of output classes across different tasks, a challenge that LLMs sidestep?
>
> We are enthusiastic about this direction and will add a dedicated discussion in our Future Work section to highlight the importance of this comparison.
>
>
> ---

---

> ### Author Response · Authors · 2025-11-19
>
> >3.*There is much heterogeneity in the datasets, as the authors acknowledge, but the paper has limited exploration of the different results in these areas and what they imply for simulation. Social science is extremely broad, and it is important to differentiate topics, rigor, elicitation methods (of the humans), etc.*
>
> **Response:**
>
> Thank you for this excellent suggestion. In our revision, we will add a new, dedicated paragraph to Section 4.3 that analyzes the systematic performance patterns across different types of human behavior.
>
> * Our analysis, based on the results in Figure 3, reveals a hierarchy of simulation difficulty: Models are most successful at simulating responses to standard survey questions about stated opinions, attitudes, and self-assessments (e.g., OpinionQA, Afrobarometer). In contrast, model performance is weaker on tasks requiring the simulation of a behavioral choice, whether in economic gambles (Choices13k) or moral dilemmas (MoralMachine). This finding provides large-scale evidence for a "value-action gap" in LLMs, echoing recent work (e.g., [a]) which suggests that there exists a misalignment between LLM-generated value statements and their actions.
>
> * Additionally, models struggle most on tasks requiring them to simulate personas with beliefs or traits that deviate from the “neutral” or "helpful" and “harmless” default. This is evident in their extremely poor performance on simulating Machiavellianism (OSPsychMach), or conspiratorial beliefs (ConspiracyCorr). This aligns with a growing body of work [b,c,d] showing that LLMs tend to struggle more when simulating "atypical" or counter-normative human perspectives.
>
> [a]: Shen, Hua, Nicholas Clark, and Tanu Mitra. "Mind the Value-Action Gap: Do LLMs Act in Alignment with Their Values?." Proceedings of the 2025 Conference on Empirical Methods in Natural Language Processing. 2025.
>
> [b]: Liu, Andy, Mona Diab, and Daniel Fried. "Evaluating Large Language Model Biases in Persona-Steered Generation." Findings of the Association for Computational Linguistics ACL 2024. 2024.
>
> [c]: Kumar, Sai Adith Senthil, et al. "Can LLMs Simulate Personas with Reversed Performance? A Benchmark for Counterfactual Instruction Following." arXiv preprint arXiv:2504.06460 (2025).
>
> [d]: Yi, Zihao, et al. "Too Good to be Bad: On the Failure of LLMs to Role-Play Villains." arXiv preprint arXiv:2511.04962 (2025).
>
> ---
>
> >4.*A number of weaknesses are acknowledged by the authors but remain nonetheless, such as the reliance on multiple-choice format questions—understandable, but limits applicability to other research formats.*
>
> **Response:**
>
> Thank you very much for raising this point. Our focus on the multiple-choice question (MCQ) format is grounded in its widespread use in the social and behavioral sciences, from large-scale surveys like the General Social Survey to classical psychology experiments like the Trolley Problem.
>
> We see an analogy to the role of foundational benchmarks in other areas. For instance, MMLU uses multiple-choice questions as a standardized instrument to measure the broader capability of "multitask knowledge and reasoning." Few would argue that reasoning is only the ability to answer MCQs; rather, the format provides a way to quantify and track progress on a complex, otherwise hard-to-measure capability. Similarly, SimBench uses distributional prediction on multiple-choice tasks as a rigorous instrument to measure the fundamental capability of human behavior simulation.
>
> While future benchmarks will undoubtedly explore more complex, open-ended formats, such work faces a significant bottleneck: the lack of large-scale, structured ground-truth data for group-level free-text or interactive behaviors. By focusing on a well-defined and widely used format, SimBench provides the essential foundation needed to establish baselines and move the field toward a reproducible science of LLM simulation.

---

> ### Author Response · Authors · 2025-11-19
>
> >5.*I read the "Training Data Overlap" section, but did the authors empirically address in any way the possibility of models, especially larger and newer models, having the datasets (or the papers resulting from those datasets) in the training data? It would be useful to include tests on unpublished data, as did Hewitt and Ashokkumar et al. (2024; already cited in the paper), or at least foreground this limitation in the main text.*
>
> **Response:**
>
> Thank you, and yes, we agree that data contamination remains a risk, especially since we cannot directly audit the training data of the closed and open-weight models that we evaluate.
>
> Unfortunately, we believe that the proposed solution—testing on truly unpublished data—is becoming logistically infeasible for the types of large-scale datasets used in SimBench. The core challenge is the rapidly shrinking gap between social science data publication and model knowledge cutoffs. Large-scale survey / experiments often have a lead time of over a year before public release. Meanwhile, the knowledge cutoffs for our top-performing models are increasingly recent; for instance, GPT-4.1's is June 2024 [a] and Claude 3.7 Sonnet's is November 2024 [b]. This "data lag" makes it practically impossible to find a major, contemporary social science dataset that is guaranteed to be unseen by the latest generation of models.
>
>
> As our "Training Data Overlap" section details, the zero-shot aggregation nature of our task, combined with the clear performance ceilings we observe, strongly suggests that our results reflect true simulation ability.
>
> We agree this is a crucial issue. In our revision, we will foreground our discussion of data contamination in the main text.
>
> [a]: https://openai.com/index/gpt-4-1/
>
> [b]: https://assets.anthropic.com/m/785e231869ea8b3b/original/claude-3-7-sonnet-system-card.pdf
>
>
> >6. *The analyses in the paper focus on the SimBench dataset as a whole, but with its diversity, what differences are there in performance across it? I ask in part because the target for simulations is often new studies where the results are highly uncertain. Ideally, one could examine performance on the parts of SimBench that are the best proxies for this (e.g., studies where humans were surprised by the results, studies on less-well-studied topics).*
>
> **Response:**
>
> Thank you, this is a great point, which will help make SimBench more practically useful. While we cannot know with certainty how models will perform on any specific new study, SimBench allows us to anticipate when simulations of uncertain outcomes are likely to be more or less reliable, based on our expanded analysis of per-dataset performance (as discussed in our response to your third point): models are more accurate at simulating stated opinions and judgments than decisions. Therefore, for a new study aiming to forecast public opinion, simulations may be more trustworthy. Additionally, when the simulation target is counter-normative, simulation accuracy is likely going to be less reliable.
>
> Crucially, these insights are framed within the context of group-level simulation, and do not necessarily extend to individual-level prediction, which presents a different set of challenges. In conclusion, SimBench provides the first large-scale map of this reliability landscape. For researchers targeting new studies, our findings are clear: simulation fidelity is not monolithic but is a function of the task's nature.

---

> ### Author Response · Authors · 2025-11-24
>
> Please let us know if there are any remaining questions or concerns. We would be happy to address them before the discussion period closes.

---

### Official Review · Reviewer_HwhX · 2025-11-01

**Soundness:** 3
**Presentation:** 4
**Contribution:** 3
**Rating:** 6
**Confidence:** 4

**Summary:**

This paper establishes a benchmark by collecting 20 existing datasets to evaluate how LLMs simulate human behavior, particularly in terms of group-level responses. The authors also conduct extensive simulations based on it and perform detailed analyses.  Although there are some issues and limitations, this work can provide new standards and references for future research.

**Strengths:**

1. The paper establishes a new benchmark based on existing datasets and performs a large number of simulations and analyses, with a lot of additional information in the appendix.

2. The structure of the paper is well-organized, and the authors provide detailed discussions of the different RQs. The figures and tables are presented in an appropriate and clear manner.

3. As the authors point out,  most of current evaluations for human behavior simulation are fragmented, and their work could serve as a useful reference for subsequent studies.

**Weaknesses:**

1. From a mathematical perspective, the score S in Eq. (1) is not a well-defined concept because if P is entirely random (i.e., identical to U), the value of S does not exist. The variable i in Eq. (1) also lacks the definition and explanation.

2. The authors assess the simulations of different LLMs based on only their own proposed metric S, without using other individual or group-level metrics. If the authors intend to demonstrate that their metric provides new insights, they need to compare it with some existing metrics. Appendix F compares TVD with other metrics, but the one used in this paper is not TVD.

3. More datasets don't always lead to more representative results. There are still more factors to consider.
(1) Survey responses are collected through various ways, such as online, phone, and face-to-face interviews, etc. Theoretically, the same person could answer the same question differently under different circumstances. This paper does not analyze the dataset based on this criteria.
(2) Do the authors consider the different quality levels of these 20 datasets? And the information loss resulting from different question normalizations in different datasets?

4. Certain conclusions might need to be modified or reviewed more carefully. For example, the conclusion "there is a clear log-linear scaling law for LLM simulation ability" drawn from Figures 2 and 6 may be too strong. Different approaches to calculating or weighting the results from various datasets are likely to yield different outcomes. With too few data points, other functions could easily be used for fitting and interpretation, e.g., OLMo and Gemma.

5. Some of the theoretical analysis is incomplete and inadequate. For instance, the definition of score S is clearly related to entropy, but this is not reflected in the explanation in Appendix I. To give an example, the range of score S is quite different in high-entropy and low-entropy cases. I speculate that it is not as simple and linear as the authors attempt to describe.

Minor issues:

1. What's the difference between "MoralMachineClassic" and "MoralMachine"? Why are their results so different in Figure 3?

2. typo, L276: "Appendix 6"

**Questions:**

Please refer to the weaknesses mentioned above.

---

> ### Author Response · Authors · 2025-11-18
>
> **Thank you very much for your thoughtful review and for highlighting the paper's detailed analyses, and its value as a reference for future research. We address your specific concerns below.**
>
> ---
>
> >1.*From a mathematical perspective, the score S in Eq. (1) is not a well-defined concept because if P is entirely random (i.e., identical to U), the value of S does not exist. The variable i in Eq. (1) also lacks the definition and explanation.*
>
>
> **Response:**
>
> We thank the reviewer for this observation. The formula presented in the paper serves as the conceptual definition of our score. It articulates the underlying principle: to measure a model's predictive error relative to a uniform baseline. However, as you point out, a direct, point-wise implementation of this formula is undefined when human response is uniform.
> To address this, we simply replaced the unstable point-wise denominator with a stable, dataset-level average baseline. At dataset-level, no dataset has averages close to uniform. The exact formula used in our experiments for a single test case $i$ is
> $$S_i = 100 \left(1 - \frac{\operatorname{TVD}(P, Q)}{\operatorname{TVD}(P, U)}\right) = 100 \left( 1 - \frac{\operatorname{TVD}(P_i, Q_i)}{\operatorname{mean}_{j \in D}(\operatorname{TVD}(P_j, U_j))} \right)$$
> where:
> *   $i$ is the index of the specific test case (i.e., a single question-group pair) being scored.
> *   $P_i$ and $Q_i$ are the ground truth human and predicted model distributions for that test case $i$.
> *   $D$ is the set of all test cases belonging to the same source dataset as test case $i$ (e.g., all questions from the "Afrobarometer" dataset).
> *   $j$ is an index that iterates over every test case in the set $D$ to compute the average baseline.
> We will update Section 3 to include this explicit operational implementation and clarify the indices used. We thank the reviewer again for prompting this important clarification.
> ---
> >2. *The authors assess the simulations of different LLMs based on only their own proposed metric S, without using other individual or group-level metrics. If the authors intend to demonstrate that their metric provides new insights, they need to compare it with some existing metrics. Appendix F compares TVD with other metrics, but the one used in this paper is not TVD.*
>
> **Response:**
>
> We thank the reviewer for giving us the opportunity to clarify our metric choice. Our evaluation does not rely on a completely new metric. The SimBench score S presented in our main results is a direct, principled normalization of TVD, chosen to maximize clarity and interpretability without altering any scientific conclusions.
>
> Raw TVD scores (ranging from 0 to 1, where 0 is better) are statistically sound but lack **intuitive meaning** for a broad audience. Our S score transforms this into an intuitive 0-100 scale. The 0-100 scale has become a **standard practice** for reporting major benchmark results (e.g., MMLU, SWE-Bench, as seen in model releases from leading labs, e.g. https://openai.com/index/gpt-4-1/). Our S score provides an instant, clear interpretation: S=100 means a perfect simulation and S<=0 means the method is no better than a uniform baseline. This allows readers to immediately grasp the practical significance of a model's performance.
>
> As the reviewer astutely noted, in Appendix F, we validated our choice of metrics, and compared TVD to other metrics. Since our normalization from TVD to S is a simple **monotonic linear transformation**, the rank ordering of all models is perfectly preserved. A model that is better on raw TVD will always be better on the S score. This can be confirmed by comparing the rankings in Table 1 (S score) and Appendix F (raw TVD) - they are identical. This ensures that all of our central findings are just as if we were to use TVD directly.
>
> ---

---

> ### Author Response · Authors · 2025-11-18
>
> >3. *More datasets don't always lead to more representative results. There are still more factors to consider. (1) Survey responses are collected through various ways, such as online, phone, and face-to-face interviews, etc. Theoretically, the same person could answer the same question differently under different circumstances. This paper does not analyze the dataset based on this criteria. (2) Do the authors consider the different quality levels of these 20 datasets? And the information loss resulting from different question normalizations in different datasets?*
>
> **Response:**
>
> Thank you very much for the opportunity to clarify. Our primary goal with SimBench is to establish the first large-scale benchmark that prioritizes **breadth and diversity** across both tasks and participant populations. While we are limited by publicly available data, our curation of 20 datasets from experimental economics to moral psychology represents a significant step beyond prior, much more narrowly-focused bespoke evaluations.
>
> (1) We agree that varied collection methods introduce variance. A truly capable simulator, however, must be robust to this real-world methodological diversity. We account for this in two ways:
>
> * Our prompts explicitly model the collection context when known (e.g., "You are an Amazon Mechanical Turk worker..."), directly instructing the model to simulate respondents within that specific setting.
>
> * By including a wide array of real-world data collection contexts, SimBench provides a realistic testbed, rewarding models that can generalize across these variations.
>
> (2) Instead of imposing a single, subjective "quality" metric across disparate fields, our primary filter was curation from **highly reputable sources**, including top-tier academic publications and established research institutions (e.g., ISSP, ESS). This ensures a high standard of data integrity. Furthermore, all models are evaluated on the exact same data, ensuring our comparisons are fair.
>
> To minimize information loss, our selection criteria (§2.1) already required datasets to be in multiple-choice or ordinal response formats. This pre-selection greatly reduces the variance in normalization needed, making our unified format a robust choice for ensuring comparability across the benchmark.
>
> ---
>
> >4. *Certain conclusions might need to be modified or reviewed more carefully. For example, the conclusion "there is a clear log-linear scaling law for LLM simulation ability" drawn from Figures 2 and 6 may be too strong. Different approaches to calculating or weighting the results from various datasets are likely to yield different outcomes. With too few data points, other functions could easily be used for fitting and interpretation, e.g., OLMo and Gemma.*
>
> **Response:**
>
>
> Thank you for raising this. We will revise this claim to state that we observe a log-linear scaling trend, which does not imply general causality. Furthermore, we will add a discussion in the paper acknowledging that this trend is most substantiated  for model families with several data points (e.g., Qwen2.5, Llama-3.1) and is more speculative for those with sparser data like OLMo. We will also carefully inspect all other conclusions in the paper to ensure they are accurately qualified.
>
> ---
> >5. *Some of the theoretical analysis is incomplete and inadequate. For instance, the definition of score S is clearly related to entropy, but this is not reflected in the explanation in Appendix I. To give an example, the range of score S is quite different in high-entropy and low-entropy cases. I speculate that it is not as simple and linear as the authors attempt to describe.*
>
> **Response:**
>
> We thank the reviewer for this sharp observation. You are correct that the conceptual S score would be unstable and non-linear for high-entropy questions, which is why we designed our practical implementation to prevent this. As we clarified in our response to your first point, we normalize against a stable, dataset-level average baseline, instead of a sample-specific baseline. This makes our operationalized score a direct linear transformation of the model's raw error (TVD).
>
> ---

---

> ### Author Response · Authors · 2025-11-18
>
> >Minor1. *What's the difference between "MoralMachineClassic" and "MoralMachine"? Why are their results so different in Figure 3?*
>
> **Response:**
>
> Thank you for the question. As detailed in Appendices L.5 and L.19, MoralMachineClassic contains only 3 classic "trolley problem" scenarios, while the full MoralMachine dataset contains over 2,000 diverse and more nuanced scenarios, resulting in more varied human responses. The latter is a much harder task, explaining the large performance difference.
> We included both for reasons below:
> * MoralMachineClassic serves as a canonical baseline. It contains the three foundational trolley problems from influential prior work (Awad et al., 2020), allowing us to benchmark models against core scenarios that have defined decades of research in moral psychology.
> * MoralMachine is itself a seminal work in moral psychology that provides a rigorous test of generalization. It includes thousands of more complex and varied scenarios involving autonomous vehicles.
> ---
>
> >Minor2. *typo, L276: "Appendix 6"*
>
> **Response:**
> Thank you very much  - we will fix this in the paper.

---

> ### Author Response · Authors · 2025-11-24
>
> Please let us know if there are any remaining questions or concerns. We would be happy to address them before the discussion period closes.

---

### Official Review · Reviewer_xaiD · 2025-11-02

**Soundness:** 4
**Presentation:** 3
**Contribution:** 2
**Rating:** 4
**Confidence:** 5

**Summary:**

The paper proposes SimBench, a benchmark for how well LLMs can simulate humans. It consists of 20 datasets selected through various criteria, and the authors use this benchmark to demonstrate a series of findings such as the effects of RL, simulating demographic groups, and others.

**Strengths:**

I think the study and analysis are comprehensive and well conducted. In particular, I think additional analyses on models such as centaur, and the explanation of the trade-off between alignment and plurality are well-written and nicely included. The dataset correlation study is also well-motivated. Overall, I think this paper cleanly presents a series of findings that have been circling around the community but hasn't been articulated clearly across many tasks, so this is very nice.

**Weaknesses:**

1. I think a core weakness of the paper is the standardization procedure to multiple choice questions. LLMs are sensitive to things such as the ordering of options [1], and also the format itself [2], so only evaluating under this paradigm is quite the weakness.

2a. The simulation methods used (prompts in appendix) are quite rudimentary and there isn't much effort in using more advanced prompts such as including personal information and then aggregating, or more granular information such as interviews [3]. Evaluating models themselves may be less relevant if there are prompt wrappers that can allow the model to perform substantially better (e.g., few shot paradigms).

2b. In the prompts themselves, it is mentioned that the simulated individual is either doing xxx survey or a worker on xxx platform. I wonder if this contradicts the generality across human populations that the authors try to achieve.

3. presentation-wise, I think it would be nice to have a table of the datasets and some brief info. After a full read I wasn't able to get a good idea of all the datasets used, only the filtering criteria.

4. [4] studies simulation on the choices13k dataset and has a set of intersecting ideas on how more performative LLMs model people more rationally. I think it would be good to include in related work.

[1] Wang, et al. "Large language models are not fair evaluators." (2023).
[2] Long, et al. "Llms are biased towards output formats! systematically evaluating and mitigating output format bias of llms." (2024).
[3] Park, et al. "Generative agent simulations of 1,000 people." (2024).
[4] Liu, et al. "Large language models assume people are more rational than we really are." (2024).

**Questions:**

See weaknesses. Primary concerns are 1 and 2a. If the authors can meaningfully address these then I am happy to raise my score, as the work is promising.

---

> ### Author Response · Authors · 2025-11-17
>
> **Thank you very much for your thoughtful review and for your recognition of our study as comprehensive and well-conducted. We address your specific concerns below.**
>
> ---
> >1.*I think a core weakness of the paper is the standardization procedure to multiple choice questions. LLMs are sensitive to things such as the ordering of options [1], and also the format itself [2], so only evaluating under this paradigm is quite the weakness.*
>
> **Response:**
>
> - Thank you very much for allowing us to clarify. Our focus on the multiple-choice question (MCQ) format is grounded in its widespread use in the social and behavioral sciences, from large-scale surveys like the General Social Survey to classical psychology experiments like the Trolley Problem. Given this practical relevance, creating the first large-scale, standardized benchmark for this context was a core goal. A standardized framework was essential to enable the clear, quantifiable, and comparable evaluation needed to rigorously assess models across our 20 diverse datasets.
> - Regarding option ordering and question format, our guiding principle was to replicate the human experimental conditions as faithfully as possible. Therefore, we intentionally preserved the original option order. Humans are known to be sensitive to **option order and question format** [a,b,c], and this bias is an integral part of the ground-truth data we are simulating. To randomize or alter the order would be to test the model under different conditions than the humans, invalidating the comparison. An ideal simulator should be able to account for such human biases in its distributional prediction. Moreover, for many of our questions (e.g., those with Likert scales), the options have a meaningful semantic order that cannot be randomized without destroying the question's integrity.
>
> **Reference:**
>
> *   [a] Chan, Jason C. "Response-order effects in Likert-type scales." *Educational and Psychological Measurement* 51.3 (1991): 531-540.
> *   [b] Wildt, Albert R., and Michael B. Mazis. "Determinants of scale response: Label versus position." *Journal of Marketing Research* 15.2 (1978): 261-267.
> *   [c] Malhotra, Neil. "Completion time and response order effects in web surveys." *Public Opinion Quarterly* 72.5 (2008): 914-934.
>
> ---
>
> >2a.*The simulation methods used (prompts in appendix) are quite rudimentary and there isn't much effort in using more advanced prompts such as including personal information and then aggregating, or more granular information such as interviews [3]. Evaluating models themselves may be less relevant if there are prompt wrappers that can allow the model to perform substantially better (e.g., few shot paradigms).*
>
> **Response:**
>
> Our work tackles a different challenge than studies focused on simulating individuals.
>
> **Individual simulation** requires rich, personal data that is practically impossible to obtain for the tens of thousands of global participants across the 20 datasets in SimBench.
> **Group-level simulation**, which is our focus, is a distinct task. The system prompt is intentionally underspecified at the individual level (e.g., "You are from Germany") precisely because the ground truth is not a single person, but a diverse population. The model's task is to express this inherent individual-level uncertainty as a coherent group-level probability distribution. A successful model uses its internal world knowledge to infer the likely variance in responses within that group and outputs a distribution that reflects this pluralism.
>
> - Our evaluation aims to answer a fundamental question: how well do current frontier models  simulate human populations out-of-the-box? Complex prompt wrappers with few-shot examples would test a different, albeit valuable, skill: in-context learning. By using a zero-shot approach, we establish a clean, reproducible baseline of a model's **core** simulation capability. This is analogous to how model developers report both zero-shot and few-shot performance on benchmarks. They are distinct measures of capability. A model that requires extensive prompting to perform well is fundamentally different from one that can simulate populations inherently.
>
> - We did also explore some more complex prompting strategies in our paper. As shown in Section 4.2, applying a zero-shot Chain-of-Thought (CoT) prompt did not improve performance. This empirically supports our decision to focus on a direct prompting method as the evaluation measure in our work.
>
> - In summary, our methodology is intentionally designed to provide a clean, reproducible, and foundational measure of a model's intrinsic group-level simulation ability. Improving model performance is an important but separate goal for future work. We believe there is a need for more specialized models for simulation, and a core objective of SimBench is to provide the foundational tool to enable and measure their development.
> ---

---

> ### Author Response · Authors · 2025-11-17
>
> >2b.*In the prompts themselves, it is mentioned that the simulated individual is either doing xxx survey or a worker on xxx platform. I wonder if this contradicts the generality across human populations that the authors try to achieve.*
>
> **Response:**
>
> Our goal is to test a model's ability to simulate the **actual population** that generated the data. The responses for Choices13k, for example, were collected from MTurk workers, who have their own distinct profile, and are not representative of the general population. To ask an LLM to simulate a "general person" for this dataset would be to test it against the wrong target. A fair evaluation requires that we instruct the model to simulate the same population that we are measuring it against. In datasets where the sample is designed to be representative of a national population (e.g., Afrobarometer, European Social Survey), our prompt is correspondingly broader (e.g., "You are from Nigeria"). Here, we do not mention the data collection method, as the goal is to simulate the general population of that country.
>
> ---
>
>
>
> >3.*presentation-wise, I think it would be nice to have a table of the datasets and some brief info. After a full read I wasn't able to get a good idea of all the datasets used, only the filtering criteria.*
>
> **Response:**
>
> Thank you very much for this suggestion. We will add a compact table in Section 2 if space allows. Meanwhile, comprehensive details on all 20 datasets are available in Appendix L.
>
> ---
>
> >4.*[4] studies simulation on the choices13k dataset and has a set of intersecting ideas on how more performative LLMs model people more rationally. I think it would be good to include in related work.
>
> **Response:**
>
> We appreciate this suggestion and will include this work in the related work section.

---

> ### Comment · Reviewer_xaiD · 2025-11-20
>
> For point 1, while I do see the point that an ideal simulator should be able to simulate positional biases and therefore shuffling answers is not desired, this ideal is never going to be practical to uphold in practice. There are always going to be assumptions that are not true when considering simulated equivalence between LLMs and humans, such as how psychology experiments' stimuli are given in multimodal environments, and how there are differences between outputs such as writing/typing/clicking a mouse, and so on. I believe a stronger justification is necessary for the specific design choice that the authors employ.
>
> Furthermore, the paper states that "Question Normalization: We standardize all items into a multiple-choice format. For datasets with continuous scales (e.g., Likert scales), we map responses to discrete bins." Here, the authors break the principle to replicate the human experimental conditions as faithfully as possible. It is also likely that in performing this process, the authors mismatch humans vs. LLMs' biased tendencies by changing the format of the question to multiple choice, which is what the authors mentioned that they are trying to avoid.
>
> I consider my questions 2a and 2b addressed, though I do believe that with the approach that authors state in the response to 2b, the benchmark is limited in its generality to simulating human population rather than crowdworkers or participants.
>
> More generally, if possible I would like to request the authors to use the revision abilities granted by the conference to update the pdf, rather than promising certain changes.

---

> > ### Author Response · Authors · 2025-11-23
> >
> > Thank you very much for your response and we glad to be able to address some of your concerns. We have updated the PDF incorporating all reviewers' feedback (changes highlighted in blue).
> >
> > We wish to clarify that the binning of continuous scales applies to **only one dataset** (Jester), where the original input was a continuous slider. We maintained this exception to ensure a unified evaluation format. However, for the other 19 datasets, we utilize the exact discrete options seen by human participants. For Likert-scale questions, we use a direct mapping of existing ordinal scales (e.g., "Strongly Agree" → "(A) Strongly Agree"). Thus, for the vast majority of the benchmark, we preserve the exact response granularity and semantic structure seen by humans. Beyond fidelity, there is a fundamental psychometric reason not to shuffle. Quite a few datasets in SimBench (e.g., OSPsychBig5) are validated instruments where the specific order of ordinal options is part of the construct. Shuffling would break the scale's logic, effectively administering a different, invalid test than the one humans took.
> >
> > Regarding your point on multimodal/complex environments: We respectfully disagree that this limitation applies to SimBench. We achieved high fidelity precisely by setting the strict inclusion/exclusion criteria described in Section 2. We intentionally excluded datasets requiring complex treatment conditions or multimodal stimuli (like many used in *Psych101* in the Centaur paper). By having strict criteria, we ensure that the LLM evaluation environment **does** replicate the human presentation as faithfully as possible.

---

### Official Review · Reviewer_93sM · 2025-11-09

**Soundness:** 2
**Presentation:** 3
**Contribution:** 3
**Rating:** 4
**Confidence:** 2

**Summary:**

This paper introduces a new benchmark SimBench that aggregates 20 datasets across diverse domains (e.g., moral reasoning, economic games, global surveys) into a unified multiple-choice format intended to evaluate the ability of LLMs to simulate human behavior. With this benchmark, the authors draw several conclusions on simulation ability and certain conditions LLMs struggle more.

**Strengths:**

- The proposed benchmark provides a large-scale benchmark aggregating performance of LLMs for simulation tasks across diverse domains.
- The authors conducts empirical evaluation on a wide range of models and evaluates models in terms of many axes (e.g., how model size, inference-time compute affect their ability, what extent does simulation ability correlate with other capabilities, etc)

**Weaknesses:**

- The central claim that SimBench measures simulation ability rests on the assumption that predicting aggregate survey response distributions constitutes human behavior simulation.
- The claims of the alignment-simulation tradeoff is potentially confounded. The authors do not rigorously test different hypothesis that could also explain this observation. For instance, question difficulty correlation, dataset artifacts (correlation could be driven by specific datasets rather than a general phenomenon), prompt sensitivity. Also the analysis only uses 13 model pairs, which pairs? What was the criteria for selecting these? Also the theoretical explanation is not complete, a stronger analysis would be needed to actually explain why this particular mathematical property translates to the observed behavioral differences.
- The correlation between SimBench and MMLU-Pro (r = 0.94) is interpreted as evidence that “simulation depends on deep reasoning.” However, this correlation could reflect data overlap or shared linguistic features such as linguistic formats (both MCQ), instructional cues, answer options, etc. rather than reasoning.
- The claim of “simulating human behavior across diverse cultural, linguistic, and socioeconomic backgrounds” is not well supported by the data composition. Although the paper states that there contains international scope, the majority of included datasets are English-language and Western-centric. It states that the benchmark spans more than 130 different countries but it is unclear how many samples there actually are per country.

**Questions:**

See weaknesses above.

---

> ### Author Response · Authors · 2025-11-17
>
> **Thank you very much for your thoughtful review and for recognizing the scale of our benchmark and the breadth of our empirical evaluation. We address your specific concerns below.**
>
>
> > 1. The central claim that SimBench measures simulation ability rests on the assumption that predicting aggregate survey response distributions constitutes human behavior simulation.
>
>
> **Response:**
>
> Thank you for allowing us to clarify.
>
> - Predicting distributions of human behavior is one form of human behavior simulation, out of many. Our goal is to measure LLM simulation performance, and we posit that predicting these distributions is a **robust, falsifiable, and scalable metric** for benchmarking such capability. This is analogous to how other foundational benchmarks operate: MMLU uses MCQs as the instrument to measure the broader capability of "reasoning," but few would claim that reasoning is *only* the ability to answer MCQs. The benchmark provides the instrument, not the definition of the capability. By focusing on a quantifiable outcome, SimBench provides the first large-scale foundation to establish baselines and move the field toward a reproducible science of LLM simulation.
>
>
> - We’d also like to note that SimBench’s scope extends far beyond surveys. Our benchmark deliberately includes diverse behavioral tasks from experimental economics (Choices13k), moral psychology (MoralMachine), and abstract reasoning (NumberGame), capturing fundamental behaviors studied across the social and cognitive sciences. These are not opinion polls but measures of human choice, judgment, and reasoning in various scenarios.
>
>
> - As we acknowledge in our limitations (Appendix A), SimBench does not capture the full, interactive complexity of human behavior. Our focus on distributional group-level behaviour was a deliberate choice to establish a rigorous and measurable starting point. Therefore, our core claim is not that LLMs can "simulate behavior" because they predict distributions, but that by measuring their ability to predict these distributions across a diverse range of foundational behavioral tasks, we can begin to scientifically benchmark their simulation capabilities. We will refine the introduction to make this distinction clearer.

---

> ### Author Response · Authors · 2025-11-17
>
> > 2. The claims of the alignment-simulation tradeoff is potentially confounded. The authors do not rigorously test different hypothesis that could also explain this observation. For instance, question difficulty correlation, dataset artifacts (correlation could be driven by specific datasets rather than a general phenomenon), prompt sensitivity. Also the analysis only uses 13 model pairs, which pairs? What was the criteria for selecting these? Also the theoretical explanation is not complete, a stronger analysis would be needed to actually explain why this particular mathematical property translates to the observed behavioral differences.
>
>
> **Response:**
>
>
> We agree that a simple correlation could be confounded. That is precisely why our central evidence is not just the visualization, but a **controlled multivariate regression analysis** (detailed in Appendix I). This analysis was specifically designed to isolate the core effect from the potential confounds the reviewer raises. Our regression includes **dataset-level and model-level fixed effects**. This standard statistical technique accounts for baseline differences in question difficulty, dataset artifacts, and model capability. The interaction effect we report is what remains after controlling for these factors, and it is a highly significant one (p < 0.001). We also used a consistent prompting strategy across all models to control for prompt sensitivity.
>
>
> The 13 model pairs were chosen **systematically**. They represent all publicly available models in our evaluation set where a clear base model and a corresponding instruction-tuned counterpart of the same size and architecture could be identified, ensuring a fair comparison. The model pairs used are also listed in Table 11:
> *   **Gemma-3** (4B, 12B, 27B)
> *   **Llama-3.1** (8B, 70B)
> *   **OLMo-2** (7B, 13B, 32B)
> *   **Qwen2.5** (3B, 7B, 14B, 32B, 72B)
>
>
> This robust empirical finding is well-explained by the established view of alignment as a form of Bayesian inference (Levine, 2018; Korbak et al., 2022).
>
>
> **The Base Model as the Prior ($\pi_0$):** The base model, a product of pre-training with a next-token prediction objective, acts as the prior distribution over a vast amount of pre-training language data. Its **mass-covering** objective forces it to represent the full spectrum of observed human language, making it naturally suited to modeling pluralistic, high-entropy distributions.
>
>
> **Alignment as Bayesian Inference:** The alignment process via RLHF optimizes a KL-regularized reward objective. As formalized by Korbak et al. (2022), the optimal policy ($\pi_*$) is the product of the prior and a reward-based likelihood:
> $$
> \pi_*(x) \propto \pi_0(x) \exp\left(\frac{r(x)}{\beta}\right)
> $$
> Here, the reward function $r(x)$ defines a **likelihood** sharply peaked around a single "best" mode of response. The hyperparameter $\beta$ controls the strength of the KL penalty, directly governing the tradeoff between satisfying the reward (**mode-seeking**) and preserving the prior (**mass-covering**).
>
>
> This inference process mathematically forces the model to abandon its mass-covering prior in favor of a mode-seeking posterior. It learns to concentrate its probability mass on the single, high-reward mode, fundamentally training it to *discard* the very pluralism required to simulate high-entropy human responses. The base model describes *what people say*; the aligned model prescribes *what it should say.*

---

> ### Author Response · Authors · 2025-11-17
>
> > 3. The correlation between SimBench and MMLU-Pro (r = 0.94) is interpreted as evidence that “simulation depends on deep reasoning.” However, this correlation could reflect data overlap or shared linguistic features such as linguistic formats (both MCQ), instructional cues, answer options, etc. rather than reasoning.
>
>
> **Response:**
> - Thank you for allowing us to clarify. We agree that we cannot make causal claims about what drives simulation ability, and we explicitly interpret the correlations as *indicative* evidence. We are excited about exploring this direction in future work and will make this clearer in the paper.
>
>
> - As for potential confounders: we can largely rule out significant data overlap. SimBench questions are not about factual recall but about zero-shot distributional prediction for specific demographic groups. We do not believe that the shared MCQ format alone is a sufficient explanation. Even very weak models are perfectly capable of answering MCQ questions, so neither improvement on MMLU nor SimBench will be primarily driven by models “getting better” at following the format of MCQs per se. Further, SimBench has a distributional prediction objective whereas MMLU requires predicting the single correct answer.
>
>
> - To test this further, we analyzed benchmark correlations from the [Stanford HELM Leaderboard](https://crfm.stanford.edu/helm/capabilities/latest/#/leaderboard). We found that MMLU-Pro correlates very strongly with leading generative benchmarks like WildBench (r=0.848). If the format were the primary driver of high correlations, we would not expect MMLU-Pro to correlate so strongly with non-MCQ, generative tasks.
>
> ---
>
> > 4. The claim of “simulating human behavior across diverse cultural, linguistic, and socioeconomic backgrounds” is not well supported by the data composition. Although the paper states that there contains international scope, the majority of included datasets are English-language and Western-centric. It states that the benchmark spans more than 130 different countries but it is unclear how many samples there actually are per country.
>
>
> **Response:**
> - While we acknowledge the systemic challenge of WEIRD bias in available data, we deliberately designed SimBench to move beyond the US-centric focus of many prior work, as a robust test of simulation ability must evaluate models on their capacity to simulate a wide spectrum of global cultural, economic, and political contexts.
>
>
> - The premise that our benchmark is "majority English-language and Western-centric" is **not** supported by a quantitative analysis of its composition (full country-specific table will be included in the Appendix). Among the samples with clear country/region origins, samples from the Anglosphere West (U.S., Canada, U.K., Australia, and New Zealand) constitute only **27.9%** of SimBench. Using a broad definition of "the West" (North America, Western Europe, Australia, and New Zealand), samples from these nations account for **45.9%** of our benchmark.
>
>
> - Regarding linguistic diversity, we also want to clarify our methodology. For many constituent datasets like Afrobarometer and LatinoBarómetro, data was originally collected from participants in their local languages. Our use of official English translations was a deliberate choice to standardize the evaluation and avoid confounding simulation ability with multilingual performance. We will update our paper accordingly.

---

> ### Author Response · Authors · 2025-11-24
>
> Please let us know if there are any remaining questions or concerns. We would be happy to address them before the discussion period closes.

---

### Meta-Review · Area_Chair_UmyB · 2026-01-07

**Summary:**

The paper introduces SimBench, a benchmark designed to evaluate whether models can replicate group-level human responses across 20 diverse datasets. The results indicate that performance scales with model size, reasoning-focused models provide limited benefit, and instruction tuning improves performance on consensus questions but can reduce it on more diverse ones. The authors also replicate prior findings showing that targeting specific demographic groups lowers performance.

This work addresses a clear need for high-quality evaluations in LLM social simulations. The dataset is rigorously curated, diverse, and carefully cleaned, and the empirical findings are presented thoughtfully, with appropriate caveats and extensive documentation in the appendices. The study is both novel and compelling.

I recommend acceptance.

**Reviewer Concerns:**

The authors addressed all concerns, but no replies back from the reviewers.

**Reviewer Scores:**

Reviewer 93sM may change the score since the authors clarified and addressed the issues,

---

### Decision · Program_Chairs · 2026-01-26

Accept (Poster)